# Machine learning-based extrachromosomal DNA identification in large-scale cohorts reveals its clinical implications in cancer

Shixiang Wang [1,3], Chen-Yi Wu[1,3], Ming-Ming He[1,3], Jia-Xin Yong[1,3], Yan-Xing Chen[1], Li-Mei Qian[1], Jin-Ling Zhang[1], Zhao-Lei Zeng [1], Rui-Hua Xu [1,2,4] ✉, Feng Wang [1,4] ✉ & Qi Zhao [1,4] ✉

The clinical implications of extrachromosomal DNA (ecDNA) in cancer therapy remain largely elusive. Here, we present a comprehensive analysis of ecDNA amplification spectra and their association with clinical and molecular features in multiple cohorts comprising over 13,000 pan-cancer patients. Using our developed computational framework, GCAP, and validating it with multi-faceted approaches, we reveal a consistent pan-cancer pattern of mutual exclusivity between ecDNA amplification and microsatellite instability (MSI). In addition, we establish the role of ecDNA amplification as a risk factor and refine genomic subtypes in a cohort from 1015 colorectal cancer patients. Importantly, our investigation incorporates data from four clinical trials focused on anti-PD-1 immunotherapy, demonstrating the pivotal role of ecDNA amplification as a biomarker for guiding checkpoint blockade immunotherapy in gastrointestinal cancer. This finding represents clinical evidence linking ecDNA amplification to the effectiveness of immunotherapeutic interventions. Overall, our study provides a proof-of-concept of identifying ecDNA amplification from cancer whole-exome sequencing (WES) data, highlighting the potential of ecDNA amplification as a valuable biomarker for facilitating personalized cancer treatment.

Extrachromosomal DNA (ecDNA) was first observed in 1965[1,2], however, its critical role as an emerging cancer hallmark is only recently coming forth with the advancements in technology[3]. EcDNAs, circular DNA elements with 1 Mb size on average, exhibiting unique properties[4,5] such as cancer-specific molecules of non-Mendelian inheritance, high chromatin accessibility, and clustered mutations. Lines of effort have been made to delineate the physical structure of ecDNA, and how ecDNA could lead to oncogene amplification, active transcription as well as the ability to change tumor genomes rapidly and dynamically[6–12]. By these traits, ecDNA relieves heredity constraints, fuels tumor evolution and intra-tumor heterogeneity to make cancer more adaptable to tumor microenvironment[13,14]. As an important form of somatic focal copy number amplification, ecDNA amplification has been discovered as a prevalent event that drives aggressive tumor growth, multi-drug resistance and poor survival outcomes across widespread cancer types[15–17]. These findings suggest the potential of ecDNA as a molecular marker for cancer diagnosis and a drug target for cancer treatment. Nevertheless, clinically feasible

[1]State Key Laboratory of Oncology in South China, Guangdong Key Laboratory of Nasopharyngeal Carcinoma Diagnosis and Therapy, Guangdong Provincial Clinical Research Center for Cancer, Sun Yat-sen University Cancer Center, Guangzhou 510060, China. [2]Research Unit of Precision Diagnosis and Treatment for Gastrointestinal Cancer, Chinese Academy of Medical Sciences, Guangzhou 510060, China. [3]These authors contributed equally: Shixiang Wang, Chen-Yi Wu, Ming-Ming He, Jia-Xin Yong. [4]These authors jointly supervised this work: Rui-Hua Xu, Feng Wang, Qi Zhao. ✉e-mail: xurh@sysucc.org.cn; wangfeng@sysucc.org.cn; zhaoqi@sysucc.org.cn

approaches for identifying ecDNA amplification that rely on accessible, supportive high-throughput cancer genome data are still lacking, and the clinical relevance of ecDNA in either common heterogeneous malignances (e.g., colorectal cancer (CRC)) or in advanced therapy context (e.g., checkpoint blockade immunotherapy), remains to be elucidated.

Methods for ecDNA characterization are crucial for both fundamental and applied research on ecDNA[18]. Traditional cytogenetic techniques, such as 4′,6-diamidino-2-phenylindole (DAPI) staining and fluorescence in situ hybridization (FISH), have been employed to detect and quantify ecDNA elements[6,19]. Sequencing-based methods, including AmpliconArchitect[20], AmpliconReconstructor[21], Circle_Finder[22], and Circle-Map[23], deduce the ecDNA structures from whole-genome sequencing (WGS) data. In contrast, Circle-Seq[24] and CRISPR-CATCH[25] provide targeted ecDNA analysis with enhanced resolution. Furthermore, the ecTag method facilitates the visualization of ecDNA in alive cells through labeling the ecDNA-specific sequences with guide RNAs (gRNAs) and fluorescent markers[26].

For detecting ecDNA amplification from large-scale clinical cancer genome sequencing data, the landmark computational toolkit AmpliconArchitect[20] has been developed to reconstruct the intricate circular structure of ecDNA from WGS data in silico. Additionally, the Circle-Seq technique[24] implements a sequencing library enrichment method for circular DNA-specific enrichment that allows direct sequencing of potential circular DNA fragments. The software such as Circle_Finder[22] and Circle-Map[23] are used subsequently to identify and score the putative ecDNA junctions.

Although aforementioned techniques have been employed to understand ecDNA in multiple cancer types[5,17,27,28], there is still room for improvement in terms of cost and technical limitations. For instance, Circle-Seq is constrained by the complex experimental procedure, a small detectable ecDNA size (most below 100Kb), and the lack of clinical practice. AmpliconArchitect is designed solely for WGS data, which limits our ability to study a broader range of tumor samples from clinical cohorts that are typically sequenced by whole-exome sequencing (WES). In comparison to WGS, WES is a more cost-effective alternative that involves a trade-off between sequencing depth and genome coverage[29]. WES has been developed and optimized for use in clinical settings, particularly in prospective clinical trials, which has resulted in a wealth of WES datasets derived from patient tumor samples. Given the wealth of available WES-based data from clinical tumor samples and the capacity of WES to extract biologically relevant insights, we postulated that harnessing the potential of ecDNA amplification identification from WES data could usher in a pivotal advancement in our comprehension of ecDNA's clinical implications, particularly within varied contexts of cancer treatment.

Reconstructing the complex circular structure of ecDNA from WES raw reads is an intractable task due to the chimeric reads supporting ecDNA junction sites are generally located outside the exome[20]. However, high copy number amplification is a distinctive feature of ecDNA[18]. Previous studies[30–35] and our preliminary analysis (Supplementary Fig. 1) have demonstrated that allele-specific gene-level copy number profiles from WES, WGS, and SNP array are comparable. Therefore, instead of deciphering the entire ecDNA amplification architecture from WES[20,23], we focused exclusively on the gene-level features derived from WES.

In this study, we develop a computational framework, GCAP, that enables the identification and characterization of ecDNA amplification from clinical cancer samples using whole exome sequencing dataset. We extensively validate GCAP using WES, WGS, Circle-Seq, and SNP array data from 40 cancer cell lines and clinical specimens. With GCAP, we profile the ecDNA spectra from more than 13,000 cancer samples and subsequentially reveals its clinical implications with series of association analysis. Collectively, we demonstrate that ecDNA can improve colorectal cancer molecular subtyping scheme and serve as promising biomarkers for survival risk stratification and ICI treatment efficacy prediction in multiple cancers. Those findings can help understanding the role of ecDNA amplification in cancer pathogenesis and offer insights that could significantly impact the development of therapeutic interventions.

## Results
### Building machine learning-based computational framework and implementation for identifying extrachromosomal DNA amplification

Primarily, a gene or a tumor was classified as ecDNA+ (ecDNA positive, for a gene, ecDNA+ represents an ecDNA cargo gene[18], otherwise is not) or ecDNA- (ecDNA negative). We collected 386 tumor-normal paired WES data, along with ecDNA status of corresponding tumor WGS data identified by AmpliconArchitect[17] across 24 cancer types in the TCGA (Supplementary Fig. 2a). Allele-specific copy number profiles were yielded by ASCAT[33,35], followed by the collapsing of copy number values and ecDNA amplification status from region-level to gene-level, resulting in 7,279,221 rows of gene-level observations. Of which, 0.35% (25,724 observations) were ecDNA positive. To test if copy number and other molecular profiles could be used to predict ecDNA cargo genes, we retrieved matched gene-level expression, mutation, methylation, and fusion from TCGA and then constructed a logistic regression model for each molecular type. We selected auPRC (area under precision-recall curve) as the primary evaluation metric on gene-level prediction due to the extreme class imbalance. As expected, the copy number exhibited moderate predictive ability (auPRC = 0.595) rather than other molecular profiles (Supplementary Fig. 2b). The data might imply that ecDNAs are commonly found with elevated copy counts, a trait advantageous for sequencing-based detection strategies, given that the quantity of copies varies proportionally alongside the quantity of resulting sequencing reads. Consequently, gene copy number emerges as a practical molecular characteristic for predicting ecDNA amplifications. Taking this observation into consideration, a computational framework was devised to optimize feature engineering of copy number profiles, construct an ecDNA amplification prediction model, and assess its performance and potential applications in clinical cancer research (Fig. 1a; refer to the Supplementary file 1 for detailed methodology).

To efficiently predict ecDNA cargo genes and classify ecDNA status of a tumor, we trained a machine learning model using the XGBOOST algorithm[36] instead of logistic regression, as it offers high performance and better tuning flexibility. Group k-fold (default k is 10) splitting, cross-validation (CV) and evaluating with auPRC were incorporated to overcome our modeling and evaluation in imbalanced data set. Random search with 1000 repeats was adopted in hyper-parameter tuning (Supplementary Fig. 3; refer to the Supplementary file 1 for detailed methodology). We feature engineered eleven variables, such as copy numbers, cancer genome overall lesions (e.g., copy number alteration burden)[37–39], calibration factors (e.g., tumor purity)[40,41], and gene-specific amplicon frequency priors[17]. The importance of features and training history of the top CV model are shown (Fig. 1b, c). Total copy number, tumor ploidy and copy number alteration burden contributed about 60%, 20% and 10% to the top CV model. Training history indicated the test auPRC finally converged to 0.815 on average, suggesting a much higher performance than the model of using the gene total copy number only (Fig. 1c and Supplementary Fig. 2b). On evaluation for other performance metrics, the model exhibited high auROC (area under receiver operating characteristic curve) and specificity, while remained considerable precision and sensitivity (Supplementary Fig. 2c, d). We repeated cross-validation three times with random initial seeds, and the result confirmed that XGBOOST outperformed logistic regression in terms of both accuracy and stability in the task (Supplementary Fig. 2e).

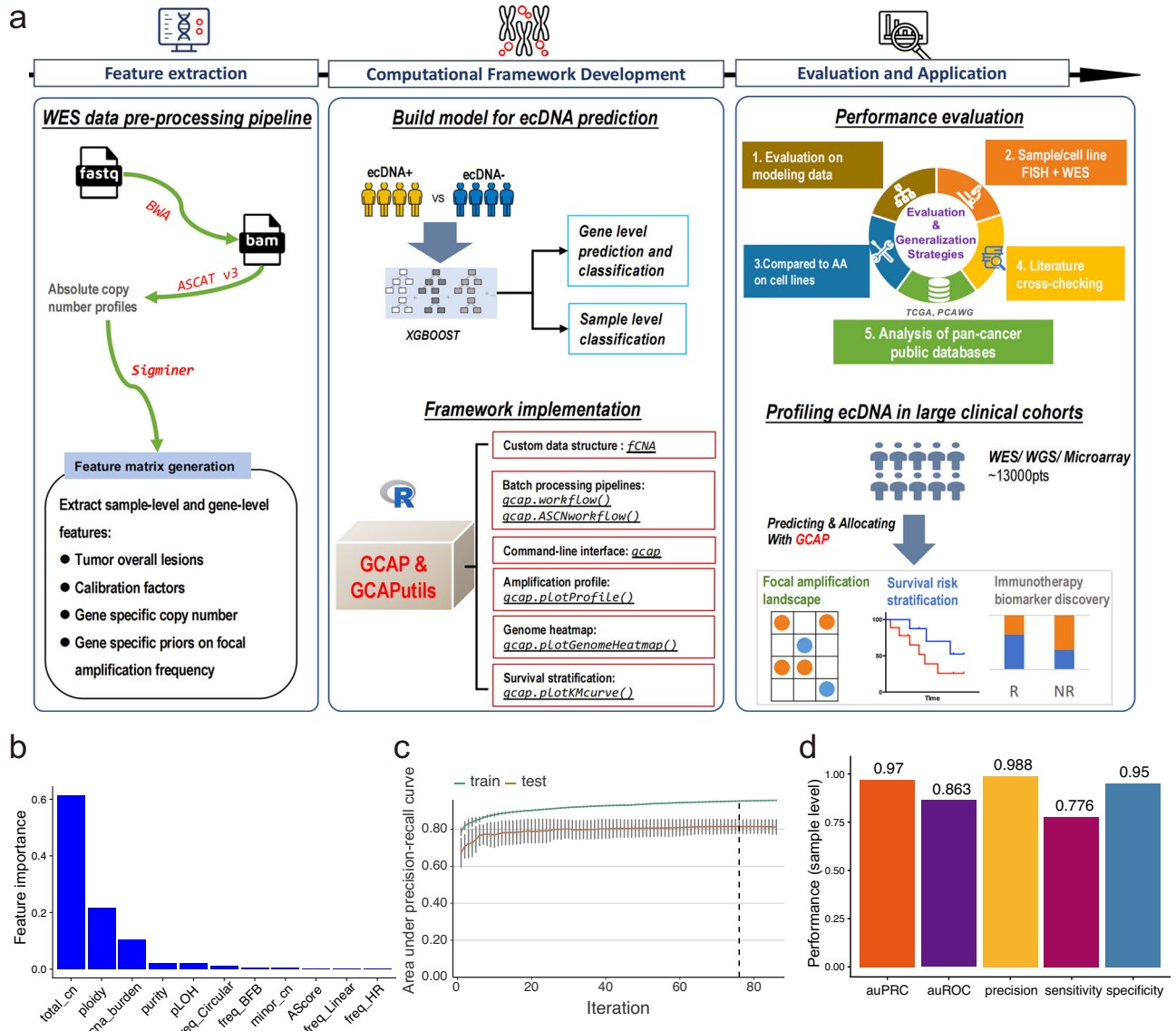

**Fig. 1 | Framework of whole-exome sequencing data-oriented cancer extra-chromosomal DNA amplification identification, evaluation, analysis, and application. a** Schematic diagram of the study. **b** Features and their importance of final constructed XGBOOST model for ecDNA cargo gene prediction. XGBOOST modeling with 11 features was repeated 1000 times independently to determine the final hyperparameters. **c** Performance estimation (auPRC, area under precision-recall curve; data are presented as mean +/- SD) for final ecDNA cargo gene pre-diction model under training and evaluation processes with stratified group k-fold cross-validation (k is 10 here). The dotted line indicates the stop iteration by early stopping approach (the performance does not improve for 10 rounds afterwards). Tumor sample size $n = 386$. **d** Performance scores auPRC, auROC (area under receiver operating characteristic curve), precision, sensitivity, and specificity of sample level ecDNA amplification identification. Source data are provided as a Source Data file. XGBOOST, eXtreme Gradient Boosting. total_cn, total copy number. minor_cn, copy number of minor allele. cna_burden, copy number alteration burden. pLOH genome percentage with loss of heterozygosity. AScore aneuploidy score.

The comparisons between the XGBOOST model with 11 predictive features (XGB11) and models with additional features, including recently cataloged copy number signatures[35], clinical factors, and cancer types, showed similar performances (Supplementary Fig. 2e). The results indicated that XGB11 is a superior model compared to the other investigated models. This was due to its lower number of fea-tures and its ability to perform well across all types of cancer. There-fore, XGB11 was likely an unbiased approach for predicting ecDNA amplification in any cancer type of interest. As the XGBOOST algo-rithm can automatically handle missing features and the main con-tributors (total copy number, tumor ploidy, and copy number alteration burden) can be obtained from absolute copy number calling[30-33,42], the XGB11 model was not further simplified. In a clinical cohort analysis, ecDNA amplification profiling is typically conducted at the sample level to investigate its potential links to available clin-icopathological features and outcome events. We then measured the performance scores of XGB11 for ecDNA amplification detection at the sample level, resulting in high metrics with auPRC (0.970), auROC (0.863), precision (0.988), sensitivity (0.776), and specificity (0.950) (Fig. 1d).

Considering a significant percentage of ecDNA negative cancers have little copy number amplification[43], overall lower genome ampli-fication level in ecDNA negative cancers probably confounds the interpretation of differential results between ecDNA positive and ecDNA negative in a cohort analysis. To eliminate the interference, we extended a two-class system to a three-class system by focal

amplification typing (Fig. 1a). A gene or a tumor was classified as: (1) 'circular', gene resides extra-chromosomally or tumor harbors ecDNA; (2) 'noncircular', gene/tumor carries chromosomally focal amplification; (3) 'nofocal', gene/tumor has no focal copy number amplification detected. In the subsequent analyses, the two-class system and the three-class system were used by context.

Taken together, our results demonstrate the feasibility of detecting ecDNA from WES datasets and present a machine-learning model for predicting ecDNA amplification from cancer genome sequencing. To facilitate ecDNA amplification identification and data analysis, we have developed end-to-end pipelines called GCAP, which bundles a utility GCAPutils for subsequent analysis and visualization (Fig. 1a). GCAP can accept either sequencing alignment data or absolute copy number calling results as input. Both GCAP and GCAPutils are now available online as R packages or a Docker image (See code availability). For simplicity, these two packages and the constructed computational framework are henceforth referred as GCAP.

### Evaluation of GCAP for extrachromosomal DNA amplification identification

To validate the ability of GCAP in ecDNA amplification identification, we employed both cancer cell lines and clinical samples to assess its robustness and generalization.

We sequenced whole exome of two previously reported ecDNA+ cancer cell lines SNU16 (gastric cancer) and PC3 (prostate cancer)[11], and then applied GCAP for ecDNA amplification identification. The ecDNA positive statuses of SNU16 (including ecDNA cargo oncogenes *MYC* and *FGFR2*) and PC3 (including ecDNA cargo oncogene *MYC*) were successfully detected by GCAP with WES data, which were sequentially confirmed by metaphase DNA fluorescence in situ hybridization (FISH) assays on the ecDNA cargo oncogenes, WGS-based AmpliconArchitect reconstructions, as well as Circle-Seq reads mapping (Fig. 2a, b). The copy numbers of *MYC* and *FGFR2* were assessed using real-time polymerase chain reaction (qRT-PCR), revealing an exceptionally high copy number of oncogenes residing on the ecDNAs (Fig. 2c). We found high consistence of copy number estimation between WES and qRT-PCR (Fig. 2d; R = 0.98 and P = 0.00063 in PC3; R = 0.96 and P = 0.0022 in SNU16) through modeling linear regression on copy number of six selected genes located in same or different chromosome cytobands (*MYC, 8q24; TNFRSF11B, 8q24; FGFR2, 10q26; ANXA7, 10q22; SOX13, 1q32; PFKFB4, 3q21*).

Using GCAP, we identified two *ERBB2* amplified gastric cancers with ecDNA+ status from our deposited clinical tumor WES data. However, only one of them was detected as ecDNA+ based on WES data of the matched patient derived xenograft model (PDX) sample. Focus on this cancer, several genes on genome regions *6p21, 17q12*, and *17q21* were predicted to have ecDNA amplifications, including oncogenes *CCND3, ERBB2* and *JUP*. To further investigate, we performed WGS and AmpliconArchitect analysis on the PDX sample. The strong agreement between the focal amplifications obtained through GCAP and the ecDNA segment links reconstructed by AmpliconArchitect validates the reliability of GCAP, even across different sequencing platforms and methodologies (Fig. 2e). We also verified GCAP on two colorectal cancer (CRC) patients predicted as ecDNA+ with accessible DNA FISH probes, WES data and FFPE (formalin fixed paraffin embedded) samples. Specifically, the patient CRC1002 was predicted harboring *MYC* amplified ecDNA; the patient CRC1057 was predicted harboring *ERBB2* amplified ecDNA. High DNA amplification occurred in both CRC patients, while distinct amplification distribution patterns were observed for the two genes in the DNA FISH staining (Supplementary Fig. 3). The image of CRC1002 contains many nucleus regions with diffuse *MYC* amplified signals, indicating a classical pattern of extrachromosomal DNA amplification, similar to patterns observed in SNU16 staining with metaphase DNA FISH for *MYC* and *FGFR2*[11] (Fig. 2a). The image of CRC1057 largely contains

nucleus regions with locally clustered *ERBB2* amplified signals, indicating a possible gene exchanging process interplayed by ecDNAs and chromosome homogeneously staining regions (HSRs) due to selective pressure change[18,44–46], similar to patterns observed in PC3 staining with metaphase DNA FISH for *MYC* (Fig. 2b), at the same time, some diffuse signals could also be observed in a small number of cells. Furthermore, we conducted FISH on 12 additional FFPE samples obtained from 11 colorectal cancer patients (seven males, four females), all previously identified as ecDNA+ for cargo genes *MYC* or *ERBB2* by GCAP. A meticulous manual inspection consistently identified gene amplification signals as either diffuse (indicating higher confidence in association with ecDNA) or locally clustered (suggesting lower confidence in association with ecDNA; Supplementary file 2). Among the six *ERBB2*-amplified samples, which typically exhibit a pattern of locally clustered gene amplification signals in FISH, our AmpliconArchitect analysis on WGS data revealed that five displayed cyclic DNA amplicons. Of these, three were classified as ecDNA+. This further strengthens the credibility of GCAP in analyzing clinical samples.

In addition to analyzing the WGS data of SNU16 (gastric cancer) and PC3 (prostate cancer), we conducted WGS on the gastric cancer cell line SNU216. We also obtained WGS data for seven other cancer cell lines from the study by Luebeck et al. [21], representing various cancer lineages, including kidney, brain, lung, myeloid, and breast cancers. We combined these cell lines together as "cell line batch 1". As gastrointestinal (GI) cancer is the focus of our group and both gastric cancer and esophageal cancer have been reported ecDNA amplification enrichment[17], we further combined the WGS data of 20 gastric cancer cell lines and 10 esophageal cancer cell lines from CCLE project[47] as "cell line batch 2". We evaluated the performance of GCAP in a heterogeneous mixture of cancer types (i.e., "cell line batch 1"), as well as in specific cases of gastrointestinal cancer (i.e., "cell line batch 2"). This assessment was conducted by benchmarking GCAP against the results obtained from AmpliconArchitect (Supplementary Data 1). The two cell line batches showed similarly high accuracy values (0.9 in cell line batch 1 and 0.867 in cell line batch 2, respectively) and F1 scores (0.923 in cell line batch 1 and 0.905 in cell line batch 2, respectively), confirming the robustness and generalization of GCAP (Fig. 2f, g). Moreover, we conducted Circle-Seq analysis on 11 cancer cell lines predicted to have ecDNA amplifications by GCAP (Supplementary Data 1). As anticipated, all these cell lines successfully met the stringent Circle-Map[23] filtering criteria, with mitochondrial DNA sequences serving as positive controls (Supplementary Data 2). When we mapped the GCAP-predicted ecDNA cargo genes to the corresponding genomic regions with Circle-Seq sequencing depth data, we observed that these genes either located at local peaks (including oncogenes *MYC, FGFR2, CTTN, ERBB2, JUP, CCNE1, MET*, etc.) or situated within regions with sufficient coverage (Supplementary Fig. 4). Combined, these data offer direct substantiation of GCAP's validity at both the sample and gene levels.

Collectively, our validation of GCAP using WES, WGS, and Circle-Seq datasets from diverse cancer cell lines, as well as clinical samples from gastric cancer and CRC, along with confirmatory experimental and computational approaches, demonstrates the reliability and practical utility of GCAP. GCAP is a copy number profile-based approach, it not only facilitates the detection of ecDNA amplification in cancer WES data but can also be extended to analyze copy number data derived from WGS and microarray.

### Recapture of ecDNA-associated survival outcome and genomic features with pan-cancer analyses

The value of applying GCAP to clinical cohorts relies on its robustness and generalizability across large-sample datasets. Pan-cancer studies have reported multiple characteristics related to ecDNA[5,17,35,48], including poor clinical outcome, high number of DNA segments,

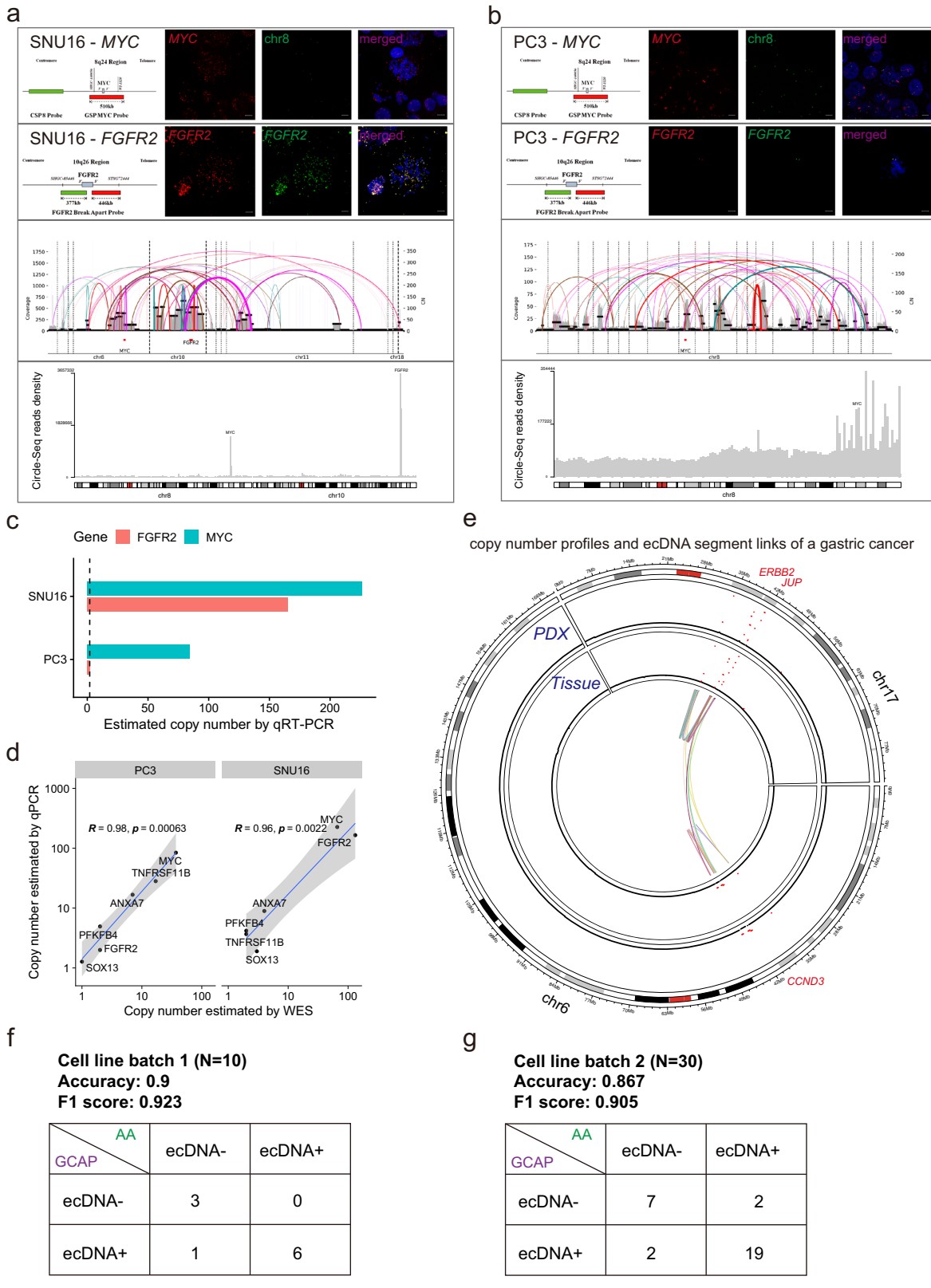

f

**Cell line batch 1 (N=10)**
**Accuracy: 0.9**
**F1 score: 0.923**

| GCAP \ AA | ecDNA- | ecDNA+ |
|---|---|---|
| ecDNA- | 3 | 0 |
| ecDNA+ | 1 | 6 |

g

**Cell line batch 2 (N=30)**
**Accuracy: 0.867**
**F1 score: 0.905**

| GCAP \ AA | ecDNA- | ecDNA+ |
|---|---|---|
| ecDNA- | 7 | 2 |
| ecDNA+ | 2 | 19 |

enriched APOBEC kataegis (a pattern of localized hypermutation), and relevant copy number signature No.8 (CN8). If GCAP could stably delineate ecDNA underlying biological profiles in large clinical cancer cohorts, then similar patterns should be observed. To test the hypothesis, we comprehensively analyzed survival outcomes and genome features by focal amplification typing on 9,699 Affymetrix SNP array-derived allele-specific copy number profiles from the TCGA

database and 2,778 WGS-derived allele-specific copy number profiles from the PCAWG database.

We firstly conducted sample-level focal amplification typing, and then compared the survival outcomes and genomic features between focal amplification subtypes including nofocal, noncircular and circular. Distinct survival outcomes were observed in the three groups (Fig. 3a–e). Circular amplification (ecDNA+) was associated with the

**Fig. 2 | Evaluation of extrachromosomal DNA amplification identification on cancer cell line genomes.** GCAP validation in two known ecDNA+ cancer cell lines **a** SNU16 and **b** PC3. The top panels show probe settings and result images of DNA metaphase FISH experiments targeting genes *MYC* and *FGFR2*. FISH result of *FGFR2* in PC3 represents a naturally negative control. The scale bar used in the figure is 10 micrometers. The middle panels show structural variant view of AmpliconArchitect (AA) reconstructions from WGS data of SNU16 and PC3. The bottom panels show Circle-Seq read density (measured as the number of reads overlapping every one-megabase window) in corresponding chromosomes. **c** *MYC* and *FGFR2* gene copy number in SNU16 and PC3 by qRT-PCR. **d** Concordance of copy number estimation by qRT-PCR and WES with six selected genes in SNU16 and PC3. Linear regression lines, point estimates of two-sided Pearson correlation coefficient test and their 95% confidence level intervals are presented. **e** Copy number profiles and extra-chromosomal DNA segment links of a gastric cancer. For better visualization, only *chr6* and *chr17*, which show ecDNA amplifications, are plotted in the Circos plot. The first and second tracks represent the total copy number of tumor tissue and patient-derived xenograft model samples. The inner track represents the extra-chromosomal DNA segment links. **f, g** Comparison between AmpliconArchitect (AA) and GCAP for extrachromosomal DNA amplification on WGS data of two cancer cell line batches. Source data are provided as a Source Data file.

poorest overall survival in both TCGA (Fig. 3a) and PCAWG (Fig. 3d). By applying a multivariable Cox regression analysis with cancer type as a covariate, we still observed patients with subtype circular held the highest risks in both overall survival (Fig. 3b, e) and progression-free survival (Fig. 3c), accounts for around 20% increased hazard compared to the patients with subtype noncircular. The APOBEC associated mutations ($P$-adj = 2.5e-29 in TCGA; $P$-adj = 1.01e-09 in PCAWG; Fig. 3f, g) and copy number signature CN8 contribution ($P$-adj = 9.98e-146 in TCGA; $P$-adj = 7.56e-135 in PCAWG; Fig. 3h, i) in subtype circular revealed a consistently elevated activity of ecDNA associated genomic alteration patterns compared to subtype noncircular. Moreover, subtype circular contained significantly more copy number segments, reflecting more genomic breakpoints than subtype noncircular ($P$-adj = 4.77e-59 in TCGA; $P$-adj = 3.21e-11 in PCAWG; Fig. 3j, k), which is consistent with the data based on AmpliconArchitect[17]. Previous studies have shown tumor purity could influence the analysis of clinical tumor samples and biological interpretation of the results[40,41]. To preclude tumor purity as the primary factor underlying the observed discrepancy between subtype circular and subtype noncircular, we examined the tumor purity. We found that tumors with focal amplification had significantly lower tumor purity than tumors with no detectable focal amplification, indicating higher inter-tumor heterogeneity. However, no difference was observed between cancer patients with circular and noncircular subtypes (Fig. 3l, m). The consistent results from the analysis of two pan-cancer databases further confirmed that GCAP is a robust cancer cohort-oriented method for identifying ecDNA amplified tumors and resolving distinct tumor heterogeneity from the focal amplification aspect.

We next analyzed genome-wide distribution and oncogene content of focal amplification. Genome-wide distribution pattern of circular amplification and noncircular amplification in TCGA (Fig. 4a) is in harmony with that in PCAWG (Fig. 4b). Frequent ecDNA cargo oncogenes *MYC*, *MYCL*, *MYCN*, *EGFR*, *ERBB2*, *FGFR1*, *MET*, *MDM2*, *MDM4*, *CCND1*, *CCNE1*, *SOX2*, *E2F3*, *CDK6*, *KRAS*, etc. were observed in both TCGA and PCAWG databases[6,8,14,15,17,27,49,50]. This demonstrates the reliability of GCAP in gene-level analysis and highlights the role of these frequently occurring oncogenes in ecDNA formation. Furthermore, similar to Kim et al. study[17], when considering DNA copy number, extra-chromosomally amplified oncogenes exhibit higher gene expression levels compared to chromosomally amplified oncogenes (Fig. 4c; F-test $P < 2.2e-16$), indicating favorable transcriptional upregulation by key properties of ecDNA other than gene dosage, such as active chromatin[7] and increased enhancer-promoter interactions[11]. According to the difference in transcriptional consequences of extrachromosomal and chromosomal focal amplification, we sought to examine the potential ecDNA associated oncogenes by multivariable linear regression modeling. Of 236 focal amplified oncogenes investigated, 104 (40%) were determined as ecDNA-associated oncogenes (Supplementary Data 3; See methods). The top 50 oncogenes with differential expression between ecDNA positive and ecDNA negative tumors are displayed (Fig. 4d).

By conducting pan-cancer analyses, we successfully reproduced the associations between ecDNA and established survival outcomes, as well as various genomic features. Moreover, we verified the efficacy of GCAP in yielding biologically significant outcomes in cohort-level analysis. Looking ahead, our next step involves the application of GCAP to real-world clinical cohorts that are of particular interest to our research group.

## ecDNA amplification is an independent risk factor in colorectal cancer

Colorectal cancer (CRC) exhibits heterogeneous outcomes and drug responses[51]. The established risk factors for CRC primarily involve lifestyle and genetics[52,53]. Previously, we established a Sun Yat-sen University Cancer Center (SYSUCC) CRC subtypes system[54], which divided the CRC into subtype HM (**H**yper**M**utated), subtype GS (**G**enome **S**table), subtype CIN-LR (**C**hromosomal **IN**stability with **L**ow survival **R**isk) and subtype CIN-HR (**C**hromosomal **IN**stability with **H**igh survival **R**isk). This classification was mainly based on the somatic single-nucleotide variations (SNVs) and copy number variations (CNVs). For ecDNA, due to the limited sample size of CRC patients in the study by Kim et al. study[17], uncertainties remain regarding the prevalence of ecDNA amplification in CRC, the correlation between ecDNA amplification and clinical characteristics of CRC, and whether focal amplification typing could provide insights into CRC heterogeneity. To address these gaps, we sought to leverage the GCAP tool to analyze a comprehensive dataset of 1,015 patients from the SYSUCC CRC cohort, which were whole-exome sequenced as part of the Changkang (Heathy Bowel) Project[54].

As a result, 164 (out of 1015, 16.2%) CRC patients were classified as circular amplified, and 246 (out of 1015, 24.2%) CRC patients were classified as noncircular amplified. We conducted a comparative analysis of gene mutations between cancer patients with circular amplifications and those without. As a result, we observed that the gene *TP53* (OR = 2.17 for 101/164 vs. 353/831, $P < 0.001$) and the pseudogene *IGHV1OR21-1* (OR = 5.24 for 7/164 vs. 7/831, $P < 0.01$) exhibited a significantly higher mutation frequency in the circular+ patient group (Supplementary Fig. 5). This finding aligns with prior research indicating that *TP53* alterations facilitate the development of ecDNA during the cancerous transformation of Barrett's esophagus[14]. To note, *IGHV1OR21-1* is a pseudogene spanning 446 bases, computational predictions (https://www.alliancegenome.org/gene/HGNC:38040) suggest its potential involvement in antigen binding activity and immunoglobulin receptor binding activity. Considering recent research indicating that pseudogenes, like other non-coding sequences, may also have regulatory functions in gene expression[55], the relationship between mutations in *IGHV1OR21-1* and colorectal cancer, as well as ecDNA, warrants further population-based analysis and experimental investigation.

Additionally, when comparing Kaplan-Meier survival curves, distinct survival outcomes were observed for both overall survival (OS) and disease-free survival (DFS) among the three focal copy number amplification (fCNA) subtypes, with the circular subtype associated with the highest risk (Fig. 5a and Supplementary Fig. 6a). Similar trends of progression-free survival (PFS) and disease-free survival (DFS) were found in the TCGA CRC cohort (Supplementary

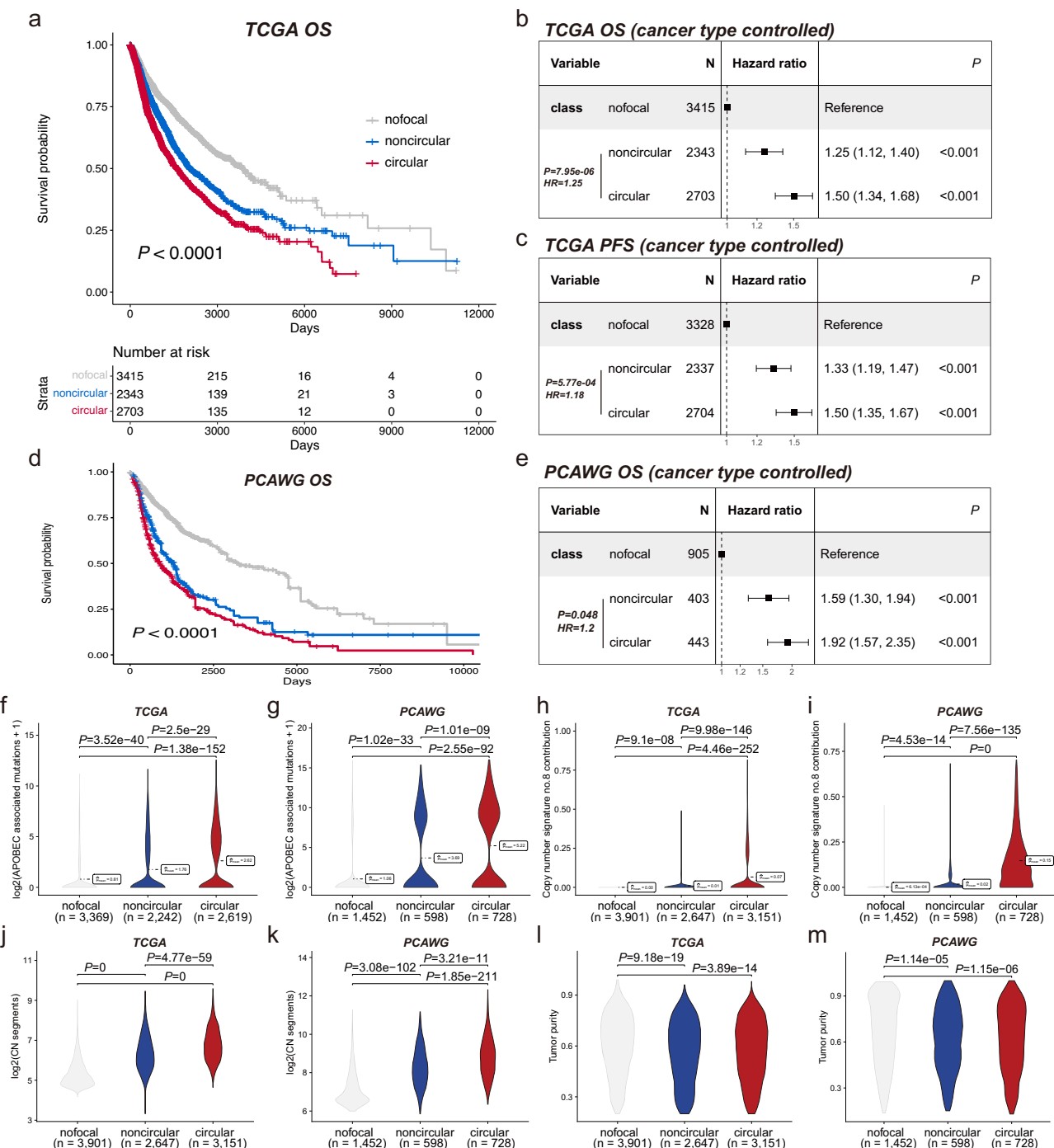

**Fig. 3 | Consistent recapture of ecDNA associated survival outcomes and genomic alteration patterns in pan-cancer databases. a** Kaplan-Meier overall survival curve comparison between different TCGA focal amplification subtypes. The exact log-rank test *P* value is 2.56e-50. Forest plots of multivariable (**b**) overall survival and (**c**) progression free survival Cox regression analysis for focal amplification subtypes with cancer type as confounding factor in TCGA. The point estimations of hazard ratio derived from Cox regression test and their corresponding 95% confidence level intervals (error bars) are presented. **d** Kaplan-Meier overall survival curve comparison between different PCAWG focal amplification subtypes. The exact log-rank test *P* value is 5.08e-38. **e** Forest plot of multivariable overall survival Cox regression analysis for focal amplification subtypes with cancer type as

confounding factor in PCAWG. The point estimations of hazard ratio derived from Cox regression test and their corresponding 95% confidence level intervals (error bars) are presented. Comparison of APOBEC associated mutations between different (**f**) TCGA and (**g**) PCAWG focal amplification subtypes. Comparison of copy number signature CN8 contributions between different **h** TCGA and **i** PCAWG focal amplification subtypes. Comparison of copy number segments between different (**j**) TCGA and (**k**) PCAWG focal amplification subtypes. Comparison of tumor purity between different (**l**) TCGA and (**m**) PCAWG focal amplification subtypes. The *P* values of comparisons in **f**–**m** were evaluated by two-sided Mann–Whitney test, with multiple comparison adjusted by FDR approach. Source data are provided as a Source Data file.

Fig. 7). By integrating fCNA subtypes with clinical characteristics, predefined SYSUCC subtypes, etc. in multivariable Cox regression analyses, we revealed that circular amplification (ecDNA+) was an independent survival risk factor in both OS (*P* = 0.01, hazard

ratio = 1.57) and DFS (*P* < 0.001, hazard ratio = 2.14) while noncircular amplification is not (Fig. 5b and Supplementary Fig. 6b). Further association analyses show that fCNA is not associated with cancer type, gender, and cigarette smoking, while significantly

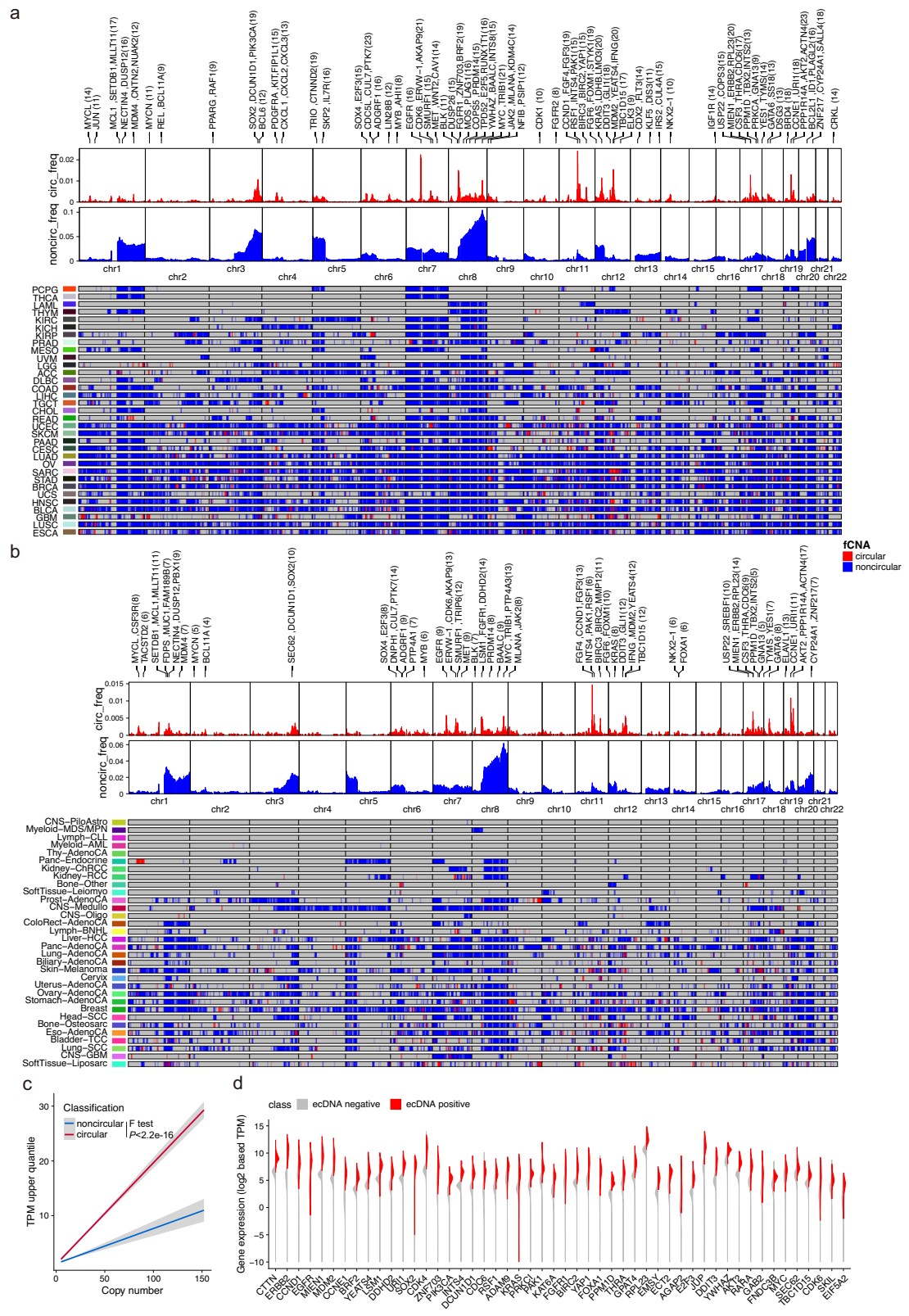

correlates with adverse features including metastasis, pathological stage, and primary tumor location (Supplementary Fig. 6c). We also found focal amplification, especially circular amplification, was mutually exclusive with MSI (microsatellite instability) in the SYSUCC CRC cohort (Supplementary Fig. 6c), and it was further confirmed with TCGA and PCAWG data (Supplementary Fig. 8a–c), although the tumor mutation burden (TMB) of focal amplified

tumors was higher than that of no focal amplified tumors (Supplementary Fig. 8d–f).

## Improvement of genomic-based molecular subtyping for colorectal cancer
Intersection of the SYSUCC subtypes and fCNA subtypes indicates subgroup GS, subgroup CIN-LR and subgroup CIN-HR have close

**Fig. 4 | Genome-wide distribution of focal amplifications and extra-chromosomal DNA associated oncogenes.** Genome-wide distribution of amplification peaks by focal amplification class in (**a**) TCGA and (**b**) PCAWG. Genomic cytobands with higher frequent circular amplification are highlighted by at most three representative oncogenes, with the count of circular amplification occurrence shown in parentheses. Here, to visualize genome-scale signals, we calculated the average signals within 1MB windows. When a spot exhibits both circular and noncircular signals, one signal may appear to be shadowed by the other. Consequently, a region within a window can only be colored in either red or blue, or grey (for none), but not simultaneously in both red and blue. To differentiate between circular and noncircular signals in such cases, readers should refer to the two bar plots on the right side of the heatmap. **c** Copy number of oncogenes versus the fold change in TPM (transcript per million) upper quartile for all oncogenes on circular

and noncircular amplification types. The fold change in TPM upper quartile is computed as the oncogene's TPM upper quartile + 1 divided by the average of TPM upper quartile + 1 for the same oncogene in all other tumor samples from the same cohort for which the oncogene was not amplified. Linear regression lines, using fold change=m × CN + b, point estimates of two-sided Pearson correlation coefficient test and their 95% confidence level intervals (in gray) are shown for each focal amplification class. This calculation is same as previously described[17]. The oncogene list was derived from Oncogene database (http://ongene.bioinfo-minzhao.org/). The two constructed linear models were compared by ANOVA analysis with F test. **d** Gene expression distribution versus extrachromosomal DNA amplification or not for top 50 ecDNA associated oncogenes. Source data are provided as a Source Data file.

---

proportion of CRC patients with circular amplification (Fig. 5c), prompting the index of chromosomal instability is not able to differentiate circular amplified tumors. Although subgroup CIN-LR enriches CRC patients with noncircular amplification, considerate proportion of CRC patients with noncircular amplification was also observed in subgroup GS and subgroup CIN-HR (Fig. 5c and Supplementary Fig. 6c). The data suggest fCNA typing may have the potential to improve existing genomic-based CRC subtypes by subcategorizing CIN tumors.

To investigate the feasibility of refining the genomic subtyping of CRC, we combined the labels of SYSUCC subtypes (HM, GS, CIN-LR, and CIN-HR) and the fCNA subtypes (nofocal, noncircular, and circular) for each cancer, then merged subgroups with close overall survival hazards into one subgroup, respectively (Fig. 5d and Supplementary Fig. 9). As hypermutation is a unique mutation pattern and mutually exclusive with focal amplification, we reserved hypermutated CRCs as an individual subtype. Finally, we built six genomic subtypes for CRC: HM (hypermutated), CIN-Mild (chromosomal instability with mild risk), CN-Quiet (copy number quiet), Non-Circ (noncircular amplification dominant), CIN-HR|Circ (either chromosomal instability with high risk or circular amplification presents), and CIN-HR&Circ (both chromosomal instability with high risk and circular amplification present). These subtypes had an increased risk compared to HM (Fig. 5d and Supplementary Fig. 9b). Comparison of Cox regression models shows that the model with combined subtypes has significantly higher C index in both OS ($P = 0.017$) and DFS ($P = 0.00067$) than the model with only SYSUCC subtypes (Supplementary Fig. 10a). Disruption of *TP53* is linked to genomic instability[14], and a strong association between *TP53* alteration status and the newly established SYSUCC genomic subtypes is evident (Fig. 5e). Notably, subtypes Non-Circ, CIN-HR|Circ, and especially CIN-HR&Circ exhibit a higher *TP53* mutation ratio than other subtypes. Additional comparative analysis of mutational signatures among the genomic subtypes (Supplementary Data 4) reveals an enrichment of ecDNA-associated APOBEC-inducing mutations and copy number signature No.8 (CN8) activity in subtypes characterized by ecDNA amplifications, with subtype CIN-HR&Circ exhibiting particularly pronounced enrichment (Fig. 5f, g). This provides independent validation of our data from previous pan-cancer analyses. Enriched mutational signatures, etiologies, and features of all six genomic subtypes are summarized in Supplementary Data 5.

To test if the genomic subtypes refinement strategy could be generalized, we applied it to the TCGA gastrointestinal cancer, a dataset had predefined molecular subtypes[56] (similar to SYSUCC subtypes above). We found the Cox regression model with molecular subtypes and fCNA subtypes has significantly higher C index in both OS ($P = 0.024$) and PFS ($P = 0.007$) than the model with only molecular subtypes (Supplementary Fig. 10b). This is in line with what we observed in the SYSUCC CRC cohort. Survival stratifications between fCNA subtypes were also observed in both GS + CIN tumors (Supplementary Fig. 10c) and CIN tumors, respectively (Supplementary

Fig. 10d). Altogether, these results suggest that focal amplification typing could be used for identifying genomic subtypes alone or extending existing genomic subtypes by supplementing more refined focal genome amplification changes.

### ecDNA amplification is predictive and prognostic for cancer immunotherapy

Comprehending the genomic correlates of response and resistance to immune checkpoint inhibitors (ICIs), either alone or in combination with other agents, can significantly benefit cancer patients by revealing biomarkers for patient stratification and resistance mechanisms for therapeutic targeting[57]. With GCAP, we are able to uncover the relations between ecDNA amplification and ICI treatment response by utilizing WES datasets within prospective clinical trials. We focused on gastrointestinal (GI) cancer and conducted a collection and analysis of four immunotherapy trials. These trials comprised two cohorts of advanced gastric cancer (AGC) treated with anti-PD1 monotherapy: SYSUCC AGC[58] cohort and SKKU AGC[59] cohort. Furthermore, we analyzed a cohort of advanced nasopharyngeal carcinoma (SYSUCC NPC cohort[60]), which also received anti-PD1 monotherapy. Additionally, we examined an advanced esophageal squamous cell carcinoma cohort (JUPITER-06 ESCC cohort[61]) that underwent either chemo+immunotherapy (toripalimab plus paclitaxel and cisplatin (TP), JS001 group) or chemo+placebo (placebo plus paclitaxel and cisplatin (TP), Placebo group) as first-line therapy. As a result, 16 (out of 55, 29.1%) and 22 (out of 55, 40.0%) advanced gastric cancers were classified as ecDNA amplified in the SKKU AGC cohort and the SYSUCC AGC cohort, respectively; 23 (out of 170, 13.5%) advanced nasopharyngeal carcinomas were classified as ecDNA amplified in the SYSUCC NPC cohort; 125 (out of 242, 51.7%) and 140 (out of 244, 57.4%) advanced esophageal squamous cell cancers from the JUPITER-06 ESCC cohort were classified as ecDNA amplified in the JS001 group and the Placebo group, respectively.

By combining the efficacy data of the SKKU AGC cohort and the SYSUCC AGC cohort, multivariable logistic regression analysis indicated that ecDNA amplification ($P = 0.02$) is a negative predictor of anti-PD1 treatment response in advanced gastric cancer, independent of the known responsive indicators PDL1 status ($P < 0.001$) and tumor mutation burden (TMB) ($P = 0.02$, Supplementary Fig. 11). We further analyzed overall survival available in the SYSUCC AGC cohort and found that the cancer patients with ecDNA amplification had worse overall survival than patients without ecDNA amplification (hazard ratio = 2.02, $P = 0.036$, Fig. 6a). Similar trend was observed in the SYSUCC NPC cohort (hazard ratio = 1.44, $P = 0.22$, Fig. 6b). Most importantly, multivariable Cox regression analysis in the SYSUCC AGC cohort (hazard ratio = 3.18, $P = 0.034$ for circular vs. nofocal; hazard ratio = 3.77, $P = 0.004$ for circular vs. noncircular) and the SYSUCC NPC cohort (hazard ratio = 1.97, $P = 0.05$ for circular vs. nofocal; hazard ratio = 1.21, $P = 0.581$ for circular vs. noncircular) showed that the circular subtype (ecDNA+) remained as a statistically significant prognostic factor for impaired survival, regardless of the presence of

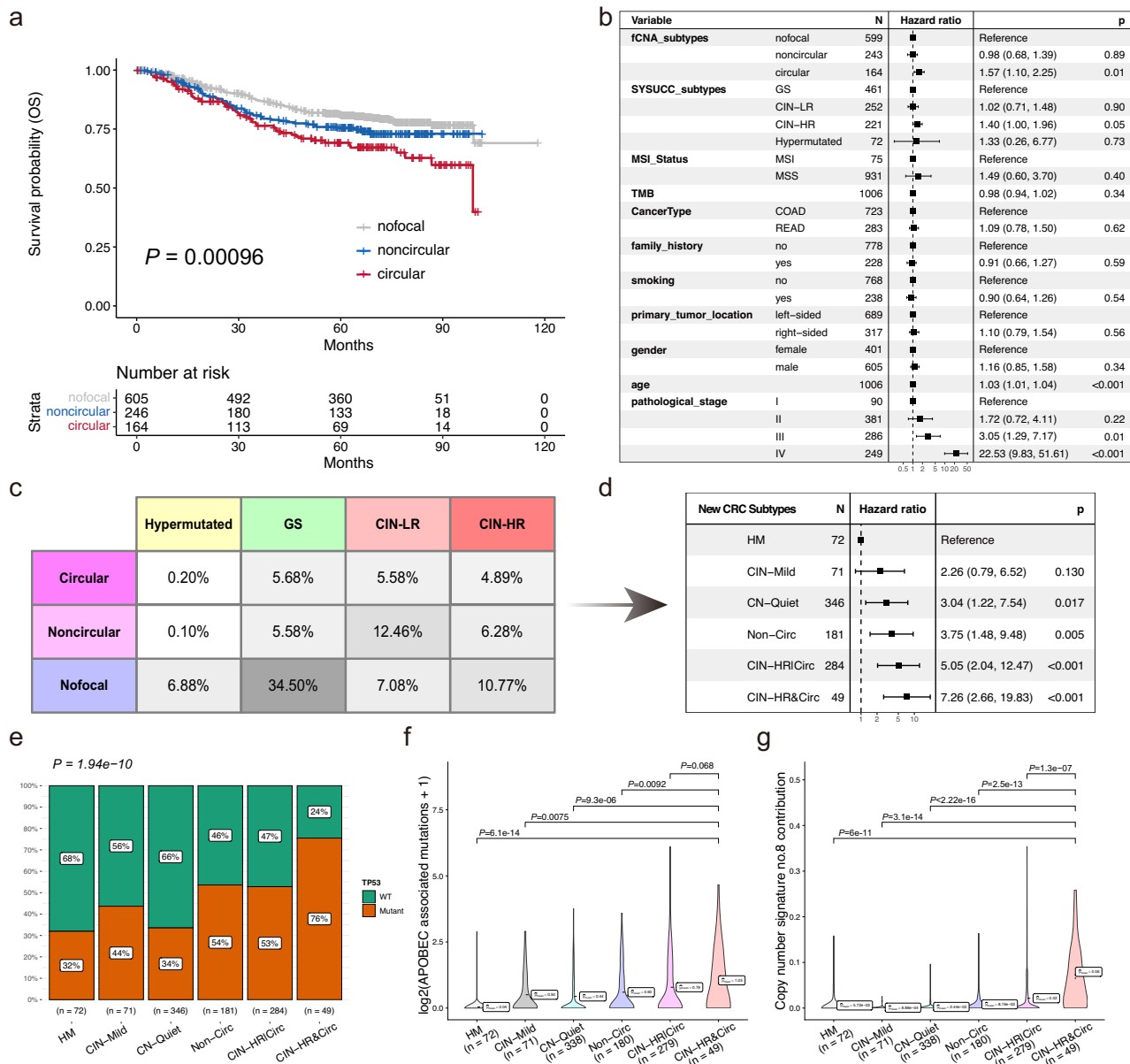

**Fig. 5 | Focal amplification typing on colorectal cancer predicts patient overall survival and yields refined genomic subtypes with distinct mutational processes. a** Kaplan-Meier overall survival curve comparison between different focal amplification subtypes. Log-rank test *P* value is shown. **b** Forest plot of multivariable overall survival Cox regression analysis for focal amplification subtypes with reported SYSUCC subtypes and other common clinical variables as confounding factors. The point estimations of hazard ratio derived from Cox regression test and their corresponding 95% confidence level intervals (error bars) are presented. **c** Combination table of existing SYSUCC genomic subtypes and focal amplification subtypes. **d** Forest plot and hazard ratios of univariable overall survival Cox regression analysis for six genomic subtypes with hypermutation group (HM) as reference. The point estimations of hazard ratio derived from Cox regression test and their corresponding 95% confidence level intervals (error bars) are presented. **e** Mutation ratio of *TP53* in the newly established SYSUCC genomic

subtypes. The *P* value estimated by Chi-squared test is reported here for showing the association between *TP53* mutation status and SYSUCC genomic subtypes. **f** Comparing APOBEC-associated mutations among the newly established SYSUCC genomic subtypes. **g** Comparing copy number signature CN8 contributions among the newly established SYSUCC genomic subtypes. The *P* values for comparisons between the CIN-HR&Circ group and other groups in **f**, **g** were evaluated by two-sided Mann–Whitney test, with multiple comparison adjusted by FDR approach. HM **H**yper**M**utated, GS **G**enome **S**table, CIN-LR **C**hromosomal **IN**stability with **L**ow survival **R**isk, CIN-HR **C**hromosomal **IN**stability with **H**igh survival **R**isk, CIN-Mild chromosomal instability with mild risk, CN-Quiet copy number quiet, Non-Circ noncircular amplification dominant, CIN-HR|Circ either chromosomal instability with high risk or circular amplification presents, CIN-HR&Circ both chromosomal instability with high risk and circular amplification present. Source data are provided as a Source Data file.

known immunotherapy biomarkers (such as PDL1 and TMB) and clinical variables (including treatment lines and liver metastasis, Fig. 6c, d). We conducted an analysis at the cytoband-level and identified two specific cytobands, *8q24* (containing oncogene *MYC*) in the SYSUCC AGC cohort and *11q13* (containing oncogenes *CCND1*, *CTTN*, etc.), where patients with ecDNA amplifications in these regions exhibited significantly worse survival outcomes compared to

patients without ecDNA amplifications in these regions (Supplementary Fig. 12). As the sample size of patients with ecDNA amplifications in these regions is relatively limited, our analysis did not reveal a statistically significant difference in survival outcomes for the other top-amplified cytobands. This observation raises the possibility of a biological role for *8q24* and *11q13* in the respective cancer types within the context of immune checkpoint monotherapy. Taken

 

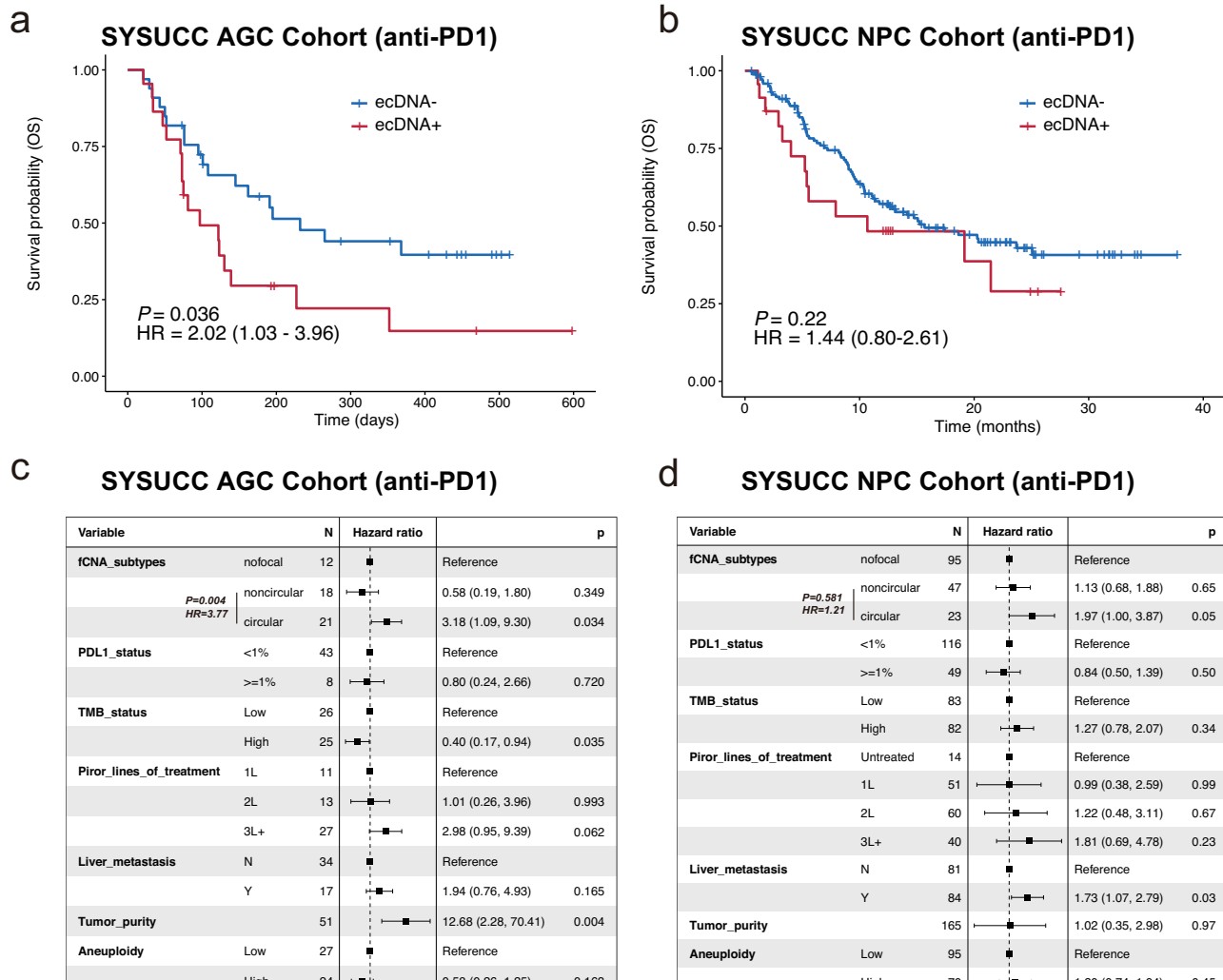

**Fig. 6 | Extrachromosomal DNA amplification is prognostic of overall survival in anti-PD1 monotherapy.** Kaplan-Meier overall survival (OS) curve comparisons between patients with ecDNA amplification and patients without ecDNA amplification in **a** the SYSUCC advanced gastric cancer cohort and (**b**) the SYSUCC nasopharyngeal carcinoma cohort. Patients in both two cancer cohorts are treated with anti-PD-1 drug toripalimab. Log-rank test *P* value and hazard ratio are shown. **c**–**d** Forest plots of multivariable overall survival Cox regression analysis for focal amplification subtypes with known immunotherapy biomarkers and other variables as control factors in (**a**) the SYSUCC nasopharyngeal carcinoma cohort and **b** the SYSUCC nasopharyngeal carcinoma cohort. The point estimations of hazard ratio derived from Cox regression test and their corresponding 95% confidence level intervals (error bars) are presented. TMB_status tumor mutation burden status. Here median as a cutoff is adopted for classifying TMB and aneuploidy into high and low groups. Source data are provided as a Source Data file.

together, these data suggest that ecDNA amplification serves as a negative predictive and prognostic biomarker for anti-PD1 monotherapy in gastrointestinal cancer.

Chemotherapy combined with PD-1 blockade has been establishing as a new standard first-line therapy for advanced esophageal squamous cell carcinoma (ESCC), there is a lack of biomarkers to aid in individualized treatment for this regimen. Recently, our group conducted an integrative analysis of WES data from treatment-naive patients with advanced ESCC enrolled in the JUPITER-06 study[62], and we identified genomic features that can distinguish between the outcomes of chemo+immunotherapy and chemo+placebo treatments.

In our extended focal amplification typing analysis, we found that the circular subtype exhibits a higher response rate (complete response + partial response) and a lower rate of progressive disease in both treatment groups (Supplementary Fig. 13a). When examining different focal amplification subtypes, we observed that patients treated with chemo+immunotherapy showed an overall survival improvement compared to those treated with chemo+placebo across all three subtypes. This improvement was statistically significant in the circular subtype (hazard ratio = 0.52, *P* = 0.0054, Supplementary Fig. 13b). Furthermore, a comparison of Kaplan-Meier curves across all subgroups revealed that patients with the circular subtype treated with chemo+immunotherapy achieved the best overall survival. When excluding this subgroup from the analysis, no significant differences were observed in the remaining subgroups (Supplementary Fig. 13c). Notably, our cytoband-level analysis identified *11q13* (containing oncogenes such as *CCND1* and *CTTN*) as the dominant focal amplification in the cohort (Supplementary Fig. 14a, b). Survival analysis specific to this cytoband showed that patients treated with chemo +immunotherapy experienced a significant overall survival improvement compared to those treated with chemo+placebo in the circular and nofocal subtypes, but not in the noncircular subtype (Supplementary Fig. 14c). Collectively, these results suggest that ecDNA amplifications, particularly those linked to *11q13*, exhibit greater sensitivity to chemo+immunotherapy compared to chemotherapy alone in advanced ESCC.

## Discussion

In this study, we introduced GCAP, a machine learning-based computational framework that provided a useful tool for exploring ecDNA amplification across multiple cancer types. We thoroughly validated the GCAP in terms of both performance evaluation and biological/clinical implications using data from various DNA sequencing platforms for varying magnitudes of multiple tumor types. Furthermore, we demonstrated the proof-of-concept of allocating ecDNA amplification from cancer WES data. We provided rich resources for studying ecDNA amplification in cancers with over 13,000 cancer focal amplification profiles generated by GCAP. With analysis of the large-scale cancer cohorts, we reported that ecDNA amplification was an independent survival risk factor for colorectal cancer, regardless of known clinical characteristics and genomic subtypes. Specifically, focal amplification typing enabled the identification of patient subgroups with distinct biological and clinical features, and optimized the existing genomic subtypes. The application of GCAP in WES datasets from clinical trials on cancer immunotherapy for advanced gastric cancer, advanced nasopharyngeal carcinoma, and advanced esophageal squamous cell carcinoma provided direct evidence of the correlation between ecDNA amplification and checkpoint blockade immunotherapy efficacy, suggesting that ecDNA amplification is a potential biomarker in gastrointestinal cancer checkpoint blockade immunotherapy. Previously, five strategies have been proposed to prolong the survival of ecDNA-related cancer patients[18], which highlights the translational potential of ecDNA, for instance, through targeting ecDNA clustering into hubs with the BET protein BRD4[11] to reduce the risk of ecDNA-driven tumor progression and therapy resistance. Our method, GCAP, enables reliable identification of ecDNA amplification from data generated by different sequencing platforms, thus enables the study of ecDNA amplification in large-scale genome datasets from both retrospective and prospective cancer studies, thereby advancing cancer diagnosis and ecDNA targeted therapy.

Interestingly, we observed a mutually exclusive relationship between microsatellite instability (MSI) and ecDNA amplification in both the SYSUCC CRC cohort and pan-cancer databases. This observation suggests that cancer cells may response to environment stress in different mechanisms depending on the initial genetic background. While MSI may confer a survival advantage by increasing mutational diversity and allowing for adaptation to different environmental conditions[63], ecDNA amplification may provide a selective advantage by carrying oncogenes or other cancer-associated genes that drive tumorigenesis and resistance to treatment[13]. Therefore, understanding the interplay between these genetic alterations and their impact on cancer phenotypes have important implications for the development of personalized therapies and the improvement of patient outcomes.

Chromosomal instability (CIN) is a hallmark of human cancer[64]. Currently, CIN level is computed by estimating the global and frequent variation patterns in cancer genomes with clustering approach[54]. Our systematic analysis indicates that focal amplification subtypes, serving as representations of localized alteration patterns, could enhance the precision of clustering-based CIN subtypes, thereby enhancing the existing genomic subtyping of SYSUCC CRC. Indeed, a more in-depth investigation into focal amplification subtypes within the CIN tumors of the TCGA gastrointestinal cancer cohort confirmed the effectiveness of focal amplifications in molecular subtyping of cancer. We anticipate that this study will encourage additional validation efforts for the integration of focal amplification typing into the planning and analysis of clinical cancer investigations involving WGS/WES. Moreover, we expect it will stimulate the creation of algorithms aimed at achieving more comprehensive characterizations of CIN levels.

Tumors with ecDNA amplification are known to be more heterogeneous and have poorer survival rates compared to those without ecDNA amplification. As such, it is not far-fetched to think that ecDNA amplification could be a useful biomarker for cancer immunotherapy[65]. However, despite this hypothesis, there has been little research exploring the connection between ecDNA amplification and cancer immunotherapy. This could be attributed to the prevalence of utilizing targeted sequencing data in immunotherapy clinical trials, whereas WGS data is currently lacking within the immunotherapy cohorts used for AmpliconArchitect analysis. In our study, through the analysis of WES data utilizing GCAP, we observed that among patients with advanced gastric cancer and advanced nasopharyngeal carcinoma featuring ecDNA amplification, those who received anti-PD1 monotherapy demonstrated a notable absence of response and experienced poor overall survival. This phenomenon may be attributed to the inherent limitations of anti-PD1 monotherapy in effectively countering the immune evasion mechanisms employed by cancer cells featuring ecDNA amplification. Interestingly, we also observed a distinct trend among advanced esophageal squamous cell carcinoma patients with ecDNA amplification. When subjected to chemo+immunotherapy, these patients demonstrated a significant improvement in overall survival compared to those receiving chemo+placebo treatment. Notably, the presence of ecDNA amplifications on *11q13* was found to be predictive of poor outcomes in the SYSUCC NPC cohort when treated with anti-PD1 monotherapy (Supplementary Fig. 12d), whereas it predicted favorable survival outcomes in the JUPITER-06 cohort undergoing anti-PD1 plus chemotherapy treatment (Supplementary Fig. 14c). This transformation from nonresponse to response may be attributed to the synergistic effects of chemotherapy-induced immunogenic cell death (ICD), therapy-induced neoantigen release, and the modulation of the immunosuppressive tumor microenvironment through anti-PD1 immunotherapy[66–68]. However, it is crucial to exercise caution in interpreting these findings. We must emphasize that the JUPITER-06 cohort represents the sole clinical trial encompassing gastrointestinal cancer patients subjected to chemo+immunotherapy combination treatment with available WES data. Nonetheless, it is essential to acknowledge the potential limitations associated with the uneven sample sizes across different focal amplification subtypes within this cohort. Taken together, it is imperative to consider the specific treatment strategy employed when evaluating the biomarker value of ecDNA amplification for patient outcomes. To gain a comprehensive understanding of the broader implications of this biomarker, additional clinical trials and data analyses involving various cancer types, particularly those characterized by immune-hot tumor microenvironments (e.g., melanoma, non-small cell lung cancer), are warranted. These endeavors will facilitate a more comprehensive assessment of the utility of ecDNA amplification as a predictive marker in diverse clinical contexts.

One caveat of applying our GCAP on clinical cancer high-throughput sequencing data is the choose of computational tools underlying copy number calling, because the reliability of prediction results depends on accurate copy number profiles, especially the accurate inference of absolute copy number for genes. To address this in GCAP workflow, we adopted ASCAT, which infers allele-specific copy number profiles under the calibration of tumor purity and ploidy with proper penalty[33]. The successful application of ASCAT in previous pan-cancer study[35] and in our work reflects the GCAP is effective in yielding proper absolute copy number profiles and infer robust focal amplification subtypes. Another caveat is the relatively low and uneven sample size of ecDNA+ tumors in different cancer types for modeling. Although we did not observe significant cancer type-specific bias introduced by uneven sample size, the predictive power might be limited by the inadequate sample size for training model. It may be improved in the future as more data become available. Another noteworthy limitation of GCAP pertains to its architectural design. Specifically, our modeling approach, primarily driven by WES data, does not facilitate the intricate reconstruction of ecDNA details. It is crucial to highlight that elevated genomic copy numbers are observed in both non-ecDNA focal amplifications (e.g., BFB) and ecDNA

amplification events. Given the pivotal role of copy number as a predictive feature, non-ecDNA focal amplifications with increased copy numbers may be prone to false positives (e.g., in cancer cell lines HCC827 and SH10TC). Simultaneously, low-copy-number ecDNAs that have not undergone significant positive selective pressures may result in false negatives (e.g., in cancer cell lines FU97 and KYSE180). The challenges faced by BFB focal amplifications in discrimination using WES data parallel those observed with WGS data. Additionally, the current version of GCAP does not encompass noncoding regions within its scope, even though noncoding elements constitute approximately two-thirds of ecDNA amplifications (Supplementary Fig. 15). Furthermore, for model training, we had to resort to using pseudo-labels derived from AmpliconArchitect due to the unavailability of extensive ground-truth data, which is an inherent limitation. Nonetheless, with the large-scale cancer cohorts we collected and multiple-faceted verifications we performed, we derived informative genomic and biological features underlying the differences in genome amplification and provided a valuable data resource of focal amplifications for future study.

Cancer genome sequencing reveals the intricate genomic complexity of tumors. Our findings demonstrate a wide range of genome-focal amplifications in various types of cancer, contributing to a better understanding of the connection between cancer genomics and clinical significance for patients. Researchers have dedicated significant time and effort to investigating and characterizing ecDNA, leading to new research directions. Future studies should investigate the integration of genomics data with other techniques like single-cell sequencing and histopathology imaging, as well as the clinical application of ecDNA in the context of targeted therapy and immunotherapy.

## Methods

The Institutional Review Board of Sun Yat-Sen University Cancer Center approved this study.

### Cell culture

The Human gastric cancer cell lines SNU16, HGC27, MKN45, NCI-N87, KATO III, MKN7, SNU216, MKN74, human esophageal cancer cell lines KYSE-410, OE19 and prostate cell line PC3, were purchased from the American Type Culture Collection (ATCC, Rockville, MD, USA). All cells except for PC3 were cultured in RPMI 1640 (GIBCO) supplemented with 10% FBS (Invitrogen) and 1% penicillin-streptomycin (Invitrogen). PC3 cells were cultured in DMEM (GIBCO) supplemented with 10% FBS and 1% penicillin-streptomycin. All cells were maintained in a humidified atmosphere of 95% air and 5% $CO_2$ at 37 °C. Cells were tested negative for mycoplasma contamination.

### Patient-derived xenograft (PDX)

All gastric cancer samples from patients were obtained surgically after informed consent. The collected tissue was immediately placed in an ice-cold DMEM culture medium with streptomycin and 5% penicillin. All mice used in this study were NOD/SCID/IL2ry$^{null}$ (NSG) female mice (6 weeks). Mice were housed under temperature-controlled, pathogen-free conditions (approximately 20 °C, 40% humidity) with a 12-h light/dark cycle. Small pieces of tissue (1–3 mm³) were directly implanted into bilateral subcutaneous pockets of NSG mice. In the initial passage of PDX, the tumor reached a volume of about 500 mm³ and then was transplanted into other mice (P2). PDX of different generations was reserved and placed in liquid nitrogen along with tissue preservation solution. Pain and distress were monitored by observing the presence of rapid weight loss, weight loss exceeding 20% of body weight, hunched posture, lethargy, lack of movement, and rapid growth of tumor masses. Mice exhibiting any of these signs were euthanized by cervical dislocation. Transplanted tumors were not to exceed a diameter of 2.0 cm or 10% of body weight as permitted by

the Institutional Ethics Committee for Clinical Research and Animal Trials of the SYSUCC.

### Fluorescent in situ hybridization (FISH)

For metaphase DNA FISH, SNU16 and PC3 cells were incubated with 50 ng/ml nocodazole (Beyotime, S1765) for 3-5 hrs to arrest them in mitosis. Then the cells were collected and resuspended in 0.075 M KCl (Sigma-Aldrich, P9541-500G) at 37 °C for 20 min, and prefixed with Carnoy's fixative ((AIDISHENG, ADS004F0, 3:1 methanol/glacial acetic acid, v/v) at RT for 10 min. The cells were centrifuged at 350 g for 5 min and fixed with Carnoy's fixative again at -20 °C for additional 30 min. After washing with fixative for three more times, cells were dropped onto microscope slides. Air-dired slides were observed under a phase contrast microscopy to avoid cell overlapping in the field and aged overnight in dark. Then slides were immersed in prewarmed 2X SSC for 5 min at 37 °C, and incubated with pepsin solution for 2 min at 37 °C. After rinsing with 2X SSC buffer, the slides were serially dehydrated in 70%, 90% and 100% ethanol each for 2 min and dried at RT. The FISH probes for MYC (LBP, F.01006) or FGFR2 (LBP, F.01197) were added onto the sample, then coverslips were applied. The samples were denatured at 85 °C for 5 min and hybridized at 37 °C in a ThermoBrite slide processing system (ThermoBrite-07J91) overnight, then washed with prewarmed 0.3%NP-40/SSC at 72 °C for 2 min and 0.1%NP-40/2XSSC for 30 sec at RT. Then the slides were immersed in 70%, 90% and 100% ethanol each for 2 min at RT and stained with DAPI and mounted. Images were acquired on a Zeiss LSM 980 confocal microscope with a 63X oil lens.

For formalin-fixed paraffin embedded tissue sections of human CRC samples, slides were deparaffinized with xylene and serially dehydrated with ethanol, then incubated in EDTA-Trisbase buffer for 20 min at 95–99 °C and treated with pepsin solution for 8-10 min at 37 °C. After rinsing with 2X SSC buffer, the slides were fixed in 10% neutral buffered formalin, and dehydrated again in 70%, 90%, 100% ethanol each for 2 min and dried at RT. The FISH probes for MYC (LBP, F.01006) or ERBB2 (LBP, F.01359) were added onto the sample with the addition of a coverslip. Samples were denatured at 85 °C for 5 min and hybridized at 37 °C overnight in a ThermoBrite system (ThermoBrite-07J91), then washed with 2XSSC then 0.1%NP-40/2XSSC for 5 min. The slides were immersed in 70%, 90% and 100% ethanol each for 2 min, and DAPI was applied to samples for 10 min. Images were acquired on a Zeiss LSM 980 confocal microscope with a 63X oil lens or by Slide scanner (3DHISTECH). The patient tissue samples utilized in this study were acquired with informed consent and received approval from the institutional review boards at the Sun Yat-sen University Cancer Center, Guangzhou, P. R. China.

### DNA copy number evaluation by quantitative Real-time PCR (qRT-PCR)

Absolute quantification is performed by plotting a standard curve with known concentrations of standards to extrapolate the amount of samples. The standards are diluted to a series of concentrations, used as templates for PCR reactions, and the standard curve is plotted using the logarithm of the copy number of the standards as the horizontal coordinate and the measured CT value as the vertical coordinate. For the quantification of the unknown DNA sample, the gene copy number can be calculated based on the corresponding CT value of the sample.

Genomic DNA was extracted from SNU16 and PC3 cells using TIANamp Genomic DNA Kit (Tiangen, Cat#DP304-03) following the manufacturer's instructions. Then we used qRT-PCR to determine the copy number for genes of interest: ANXA7, FGFR2, MYC, PFKFB4, SOX13 and TNFRSF11B. The quantitative PCR was performed on the Light-Cycler 480 instrument (Roche Diagnostics, Switzerland) using the following conditions: an initial denaturation at 95 °C for 2 min, followed by 40 cycles of 95 °C for 15 sec, 60 °C for 15 s and 72 °C for 20 s, then 95 °C for 5 s and 60 °C for 1 min. ANXA7 was amplified using the

forward primer 5′-CTCACGGCTCACACTGCTTA and the reverse primer 5′-GGGAGACTAGGGACCGATGA, *FGFR2* was amplified using the forward primer 5′-GCTTTGAGGATGTCTGGGCT and the reverse primer 5′-AGTCCCGCCATTGAAGTCAG, *MYC* was amplified using the forward primer 5′-GCGAGGATGTGTCCGATTCT and the reverse primer 5′-CCCTTCGCACTCAATACGGA, *PFKFB4* was amplified using the forward primer 5′-GCTCTAGTGGGAGGAGGTCA and the reverse primer 5′-TTTCTCCGCTGCTCATGTGT, *SOX13* was amplified using the forward primer 5′-TGTTGGGAGGCTAATGGCTG and the reverse primer 5′-GCCTAGCTCAACCGACACTT, and for *TNFRSF11B*, forward primers 5′-CTGATACAATCTGAAGGCCATCCC and reverse primer 5′-GAGGACCTCTTCCCATGCACT. Each sample was assayed in triplicate.

### DNA extraction and whole-exome/whole-genome sequencing

Genomic DNA was extracted from blood, tumor, and patient-derived xenograft samples of gastric cancer patients, frozen tissue sections of colorectal cancers patients, as well as from cancer cell lines PC3, SNU16 and SNU216, using the DNeasy Blood & Tissue Kits (QIAGEN) or the TIANamp Genomic DNA Kit (Tiangen, DP304). Extracted DNA was then quantified by Qubit 3.0 (Thermo Fisher Scientific, Inc., Waltham, MA, USA), in accordance with manufacturer's instructions. DNA was sheared using enzyme dsDNA Fragmentase (New England BioLabs, Inc., Ipswich, MA, USA). The fragmented genomic DNA underwent end-repairing, A-tailing and ligation, and then was sequentially completed with indexed adapters, followed by size selection using Agencourt AMPure XP beads (Beckman Coulter Inc., Brea, CA, USA). The DNA fragments were used for library construction with the KAPA Library Preparation kit (Kapa Biosystems, Inc., Wilmington, MA, USA) according to the manufacturer's protocol. Seven to eight polymerase chain reaction (PCR) cycles, depending on the amount of DNA used, were performed on pre-capture ligation-mediated PCR (Pre-LM-PCR) Oligos (Kapa Biosystems, Inc.) in 50 µL reactions. The whole-exome sequencing (WES) was performed using Agilent SureSelect Human All Exon V6 capture kit on the Illumina NovaSeq 6000 system (150 bp paired end) according to the manufacturer's instructions, with an average depth of 200X. The whole-genome sequencing (WGS) was performed on the Illumina NovaSeq 6000 system (150 bp paired end) according to the manufacturer's instructions, with an average depth of 30X or 10X.

### Circular DNA extraction, sequencing and analysis

Circular DNA was harvested from the human cancer cell lines described above using Monarch® HMW DNA Extraction Kit (NEB, #T3050). The linear chromosomal DNA was digested by plasmid-safe ATP-dependent DNase (Lucigen, E3101K) with 25 mM ATP and provided reaction buffer. The reaction was incubated at 37 °C overnight, and the DNase was heat-inactivated by incubating at 70 °C for 30 min. Circular DNA was next purified using an eccDNA purification kit (CAT#:220501-50), then amplified using the REPLIg Midi Kit (QIAGEN, 150043) according to the manufacturer's instructions. Amplified circular DNA was measured with a Qubit 3.0 Fluorometer (Thermo Fisher Scientific) and sheared to an average fragment size of 150–200 bp. Libraries were prepared using the NEBNext Ultra DNA Library Kit for Illumina according to the manufacturer's protocol (New England Biolabs) and sequenced with 2×150-bp paired-end reads on the Illumina NovaSeq 6000 system according to the manufacturer's instructions, with an average data volume of 30 G reads.

To detect extrachromosomal circular DNA regions, we employed the Circle-Map Realign algorithm[23] and applied the following stringent filtering criteria: CircleScore > 50, the presence of more than one discordant read, at least four split reads, a mean coverage exceeding 4, a coverage continuity below 0.1 (indicating extensive read coverage across the entire circular DNA region, indicative of high-quality identification), and a region length exceeding 10 kb. Detailed data are available in Supplementary Data 2. Notably, our approach differs from

the standard Circle-Seq procedure for ecDNA detection as we retained mitochondrial DNA sequences in our analysis, serving as internal positive controls for each cancer cell line.

### Allele specific copy number calling and feature extraction

ASCAT v3[33] (https://github.com/VanLoo-lab/ascat) was used to generate the allele specific copy number profiles for collected tumor-normal paired whole-exome sequencing data of cancer patients from TCGA, Changkang project and anti-PD1 clinical cohorts. More specifically, based on ASCAT instruction for processing high-throughput sequencing data, (1) ascat.prepareHTS function was used to derive logR and BAF from high-throughput sequencing data; (2) GC content correction and replication timing correction were applied; (3) penalty was set to 70 in ascat.aspcf function; (4) gamma was set to 1 in ascat.runASCAT function. By applying Sigminer[34] on the allele specific copy number profiles, seven sample-level or gene-level features include total_cn (total copy number), minor_cn (minor copy number), tumor purity[40,41], tumor ploidy, pLOH (genome percentage with LOH[35]), AScore (aneuploidy score[38,39]) and cna_burden (copy number alteration burden[37]) were yielded for each gene. Meantime, four amplicon types (Circular (extrachromosomal DNA), BFB (breakage-fusion-bridge), HR (heavily rearrangement), Linear) and corresponding amplicon regions detected by AmpliconArchitect on WGS of 3,212 cancer patients[17] were obtained for generating frequency of being any amplicon type for each gene, resulting in four features named freq_Circular, freq_BFB, freq_HR, freq_Linear. The four gene-level priors were then combined to seven features above by joining by gene identifier, resulting in 11 features as a basic feature set for gene-level prediction modeling.

### EcDNA identification with AmpliconArchitect, tumor only copy number calling and feature extraction

For collected tumor only whole genome sequencing FASTQ data of 40 cancer cell lines (Supplementary Data 1), the bioinformatics best-practice analysis pipeline[69] for the identification of ecDNAs with AmpliconArchitect approach (https://nf-co.re/circdna, v1.0.2) was directly applied and identification results were outputted. As AmpliconArchitect uses CNVkit[32] (https://github.com/etal/cnvkit, RRID:SCR_021917) for generating seeding regions from copy number profiles, the results of CNVkit were also collected for further thresholding with cell line specific gender and fixed tumor purity 1 to get absolute copy number profiles. Afterwards, 11 features described in section above were derived. Note that NA values were set to minor_cn and pLOH as they could be obtained or calculated from total copy number profiles. Similarly, self-generated tumor only whole-exome sequencing data of cancer cell line PC3 and SNU16 were analyzed by CNVkit batch command to get copy number segmentation results, and then re-called with cell line specific gender and fixed tumor purity 1 to get absolute copy number profiles. Subsequently, 11 gene-level features were derived.

### GCAP modeling and implementation

For this part, please refer to the Supplementary file 1 for detailed methodology.

### Association analysis between ecDNA and oncogenes

To explore oncogenes associated with extrachromosomal DNA amplification, oncogene list was first derived from Oncogene database[70] (http://ongene.bioinfo-minzhao.org/), then a multivariable linear regression model on gene-level data was built to discover ecDNA associated oncogenes. In brief, a multivariable linear regression model for each gene with the following formula:

$$TPM = a \times CN + b \times Circular + c \times CancerType + d \times TumorPurity + e$$

The formula was used to check if a gene is amplified on ecDNA ("circular") could significantly influence its own transcription level

under the adjustment of confounding factors including gene copy number, tumor cancer type and tumor purity. Multiple hypothesis testing was applied to results of all genes with FDR method. An ecDNA associated oncogene was then determined if the FDR value of coefficient b for this gene is less than 0.05.

## Mutational signature analysis

Mutational signatures were identified using R package Sigminer v2.2.0[34] (https://github.com/ShixiangWang/sigminer) with reference fitting approach. Contribution of SBS signatures was refitted with COSMIC SBS signature database v3[71] (https://cancer.sanger.ac.uk/signatures/sbs/) as reference. Contribution of copy number signatures was refitted with previously reported allele-specific copy number signatures[35] as reference. APOBEC associated mutations were estimated by summation of absolute activity of APOBEC mutational signature SBS2 and SBS13. Hypermutated signature is defined as any of the following signatures based on the annotation information from the COSMIC database: SBS6, SBS9, SBS10, SBS14, SBS15, SBS20, SBS21, SBS26 and SBS44.

## Statistical analysis

Continuous data between two or multiple groups were compared using a Mann–Whitney test. Categorical data between two or multiple groups were compared using Fisher test or Chi-squared test. Correlation analysis was performed using the Pearson method. Kaplan-Meier survival curves were generated and compared using the log-rank test. The linear models were compared by ANOVA analysis with F test. Multivariate survival analysis was performed with Cox regression model. Multivariate association and response analysis were performed with logistic regression model. All reported $P$-values are two-tailed, and for all analyses, $P \leq 0.05$ is considered statistically significant, unless otherwise specified. Multiple testing $P$-values were corrected by Benjamini–Hochberg FDR method. All statistical analyses were performed using R v4.0.2.

## Reporting summary

Further information on research design is available in the Nature Portfolio Reporting Summary linked to this article.

## Data availability

Human oncogene list was obtained from the Oncogene database[70] (http://ongene.bioinfo-minzhao.org/). TCGA tumor-normal paired WES data from 386 pan-cancer samples (sample list is available in Source Data file) for gene prediction modeling were downloaded from GDC data portal with gdc-client v1.6.1 (dbGaP accession number phs000178.v9.p8). TCGA allele specific copy number profiles can be found at https://github.com/VanLoo-lab/ascat/tree/master/ReleasedData/TCGA_SNP6_hg19. PCAWG allele specific copy number profiles and survival data can be found at PCAWG Xena hub (https://pcawg.xenahubs.net). Other types of data including gene expression, mutation, survival data for TCGA are available at Pan-Cancer Atlas hub (https://pancanatlas.xenahubs.net). The raw sequence data of Changkang Project have been deposited in the Genome Sequence Archive in National Genomics Data Center, China National Center for Bioinformation / Beijing Institute of Genomics, Chinese Academy of Sciences, under accession number HRA000873. The processed clinical annotations and structured genomic dataset for Changkang Project are available at Zhao et al. [54]. and https://changkang.hapyun.com/. SRA accessions of WGS data for cancer cell lines were collected in Supplementary Data 1. WES and Circle-Seq data for cancer cell lines generated by this study were deposited in SRA BioProject under accession number PRJNA894840. The raw sequencing data related to PDX/clinical samples is deposited in the Genome Sequence Archive, under accession number HRA006537. There are no restrictions on who will be granted access to the data, and requests can be directed to the corresponding author

(zhaoqi@sysucc.org.cn). Access will be granted within approximately weeks, and there are no restrictions on how long data will be made available for. The processed data for ecDNA cargo gene modeling, GCAP and AmpliconArchitect results for cancer cell lines, TCGA, PCAWG and Changkang Project, PDX/clinical samples, etc. were deposited in Zenodo (https://doi.org/10.5281/zenodo.7272630) with open access. The remaining data are available in the Supplementary Information or Source Data file. Source data are provided with this paper.

## Code availability

Analysis code is deposited in GitHub (https://github.com/ShixiangWang/gcap-wes). Packages GCAP (R package: https://github.com/ShixiangWang/gcap; Docker container: https://github.com/ShixiangWang/gcap/pkgs/container/gcap) and GCAPutils (R package: https://github.com/ShixiangWang/gcaputils) are publicly available online for free download and academic use. To ensure proper usage of the GCAP, authorization from the corresponding author is required for any commercial use.

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

## Acknowledgements

This work was supported by the by National key R&D Program of China (2021YFA1302100 to Q.Z.), National Natural Science Foundation of China (82321003, 81930065, 82173128, 81872011, 82172861, 82303953), International Cooperation and Exchanges National Natural Science Foundation of China (82061160373 to F.W.), CAMS Innovation Fund for Medical Sciences (CIFMS) (2019-I2M-5-036 to R.H.X.), Cancer Innovative Research Program of Sun Yat-sen University Cancer Center (CIRP-SYSUCC-0004 to R.H.X.), Sun Yat-sen University clinical research 5010 program (84000-31630002), Science and Technology Program of Guangzhou (202206080011 to F.W.), Guangdong Basic and Applied Basic Research Foundation (2021A1515011743 to Q.Z.), the Fundamental Research Funds for the Central Universities, Sun Yat-sen University (84000-31620003 to F.W.), the Young Talents Program of Sun Yat-sen University Cancer Center (YTP-SYSUCC-0033 to Q.Z. and YTP-SYSUCC-0018 to F.W.), The Youth Talent Support Programme of Guangdong Provincial Association for Science and Technology (SKXRC202313 to Q.Z.) and China Postdoctoral Science Foundation (2021M703733 to S.W.). We thank Dr Tianpeng Zhang from University of Pennsylvania. We thank Prof. Haopeng Yu from West China Biomedical Big Data Center for his high-performance computer support on the initial exploration analysis. We thank Prof. Huilin Huang for the critical advice on precluding potential confounding explanation in design of focal amplification typing framework. We thank Dr. Zixin Qin for editing this manuscript. We also thank Dr. Shuaiyang Jing for the advice on metaphase DNA FISH assay.

## Author contributions

Q.Z., F.W., and S.W. conceived and designed the overall study. S.W. developed the machine learning model, focal copy number amplification analysis framework and software. S.W., M.M.H., L.M.Q. and Y.X.C. performed the data collection, data cleaning and bioinformatics analyses. C.Y.W., J.X.Y. and J.L.Z. performed sequencing and functional assays. C.Y.W., J.X.Y., M.M.H., F.W. and Z.L.Z. contributed to sample acquisition and clinical annotations. Q.Z., F.W. and R.H.X. supervised the study. S.W., Q.Z., C.Y.W., F.W., and M.M.H. wrote and revised the manuscript with contributions from all authors, and all authors reviewed and approved the final manuscript.

## Competing interests

R.H.X, Q.Z. and S.W. declare patent applications for "Gene-level focal amplification modeling and cancer typing for extrachromosomal DNA characterization" (P. R. China application serial number 202211067952.6). All other authors declare no potential competing interests.
