## [Peer Review File · Nature Communications]

Machine learning-based extrachromosomal DNA identification in large-scale cohorts reveals its clinical implications in cancerEditorial Note: Parts of this Peer Review File have been redacted as indicated to remove third-party material where no permission to publish could be obtained.

REVIEWER COMMENTS

Reviewer #1 (Remarks to the Author): expertise in ecDNA biology

In this manuscript, Wang and colleagues leverage a machine-learning based approach to infer ecDNA from whole exome sequencing (WES) data. They apply this tool to TCGA and PCAWG data sets and then focus on colorectal cancers, including data from 4 clinical trials that incorporate PD-1 immunotherapy. They show that ecDNA is associated with worse prognosis in patients, and that it is an independent risk factor in colorectal cancers, and also reveal that patients whose tumors have ecDNA have derive less benefit from anti-PD-1 therapy.

The authors have done a great deal of impressive work and many aspects of the paper are potentially very important and of high interest, including developing a way to infer ecDNA status and its contents from WES data; and showing its adverse impact on patients with colorectal cancers, including in response to anti-PD1 therapy.

There are however, significant concerns that need to be addressed, as follows:

1) Validation of the machine learning approach to make ecDNA-calls: Overall, the numbers make great sense and the comparison with Amplicon Architect data is compelling. However, the direct validation of the method is very limited. It seems essentially to boil down to FISH on 2 known cancer cell lines and 2 colorectal cancer clinical samples. While this is nice, it is not nearly adequate for validating this as an independent tool for ecDNA calling.

2) Description of the machine learning tool is very limited: The figure is nice, but in fact, it is very hard to figure out what they actually did. A more detailed treatment, even in supplement, would be very important.

3) Application to colorectal subtypes: The authors have previously published work on colorectal subtyping. However, these distinctions are not easily intuitive. A reader of this paper will now easily know what LR or HR mean in this context. The treatment feels cursory.

4) Mutational signatures component: this part is very superficial. More details or illustration is greatly needed. At present, it detracts rather than adds. it is clearly an important topic, but not well suited to cursory treatment.

5) Major concern regarding interpretation of combination therapy data related to Extended Figure 11: They attempt to show that addition of immunotherapy to chemotherapy adds benefit preferentially to patients whose tumors have ecDNA. However, this reviewer believes that this is an incorrect conclusion. Analysis of the numbers reveals that the number of patients with ecDNA in the trial is roughly double that of patients with non-circular amplification or no amplification. The curves look very similar but only the ecDNA containing tumor curve reaches significance, likely because it has double the numbers. This has truly important impact on the conclusion. Doesn't that suggest that addition of immunotherapy can help in these esophageal cancer patients in all 3 examples, even if ecDNA-containing tumors are less response to IO mononotherapy? Positing enhanced benefit in ecDNA-containing tumors by adding IO therapy may be a misleading conclusion.

Reviewer #2 (Remarks to the Author): expert in ecDNA computational analysis

Wang, et al. "Machine learning-based extrachromosomal DNA identification in large-scale cohorts reveals its clinical implications in cancer" presents a novel framework for the identification of ecDNA based on whole-exome sequencing, copy number calling and prior analysis of whole genome sequencing to predict the gene cargo of ecDNA. Wang et al. analyze the relationship of ecDNA with

gene cargo, survival, mutational signature, and anti-PD1 immunotherapy. The paper reinforces existing knowledge of ecDNA gene contents, transcriptional dynamics and association with survival, and also presents an important step forward in our understanding of how ecDNA presence affects immunotherapy.

Major issues:

1. The use of the word "wild-type" is misleading when describing non-ecDNA calls (for example in Figure 2, Supplementary Table 1, Figure 6). Rather it seems they are simply ecDNA-. Only if the copy number is measured to be normal is the label "wild-type" appropriate. Many are likely non-ecDNA focal amps, and that is certainly not a "wild-type" state.

2. In the analysis of ecDNA status and immunotherapy response (Figure 6), the analysis is performed as ecDNA+ vs. "wild-type". In reality, there are two hidden classes in the "wild-type" category; non-ecDNA focal amps, and no focal amp at all. Is there any difference in response between ecDNA+ and non-ecDNA focal amps? Specifically, does it matter that oncogene amplification is carried on ecDNA when gauging anti-PD1 response or is any mechanism of amplification sufficient to hinder response? Does it matter which genes are amplified in determining the response?

3. Discriminating breakage-fusion-bridge focal amps (BFBs) from ecDNAs is a computationally challenging task due to the high genomic copy numbers found in both events. Can the authors show that their method is able to reliably discriminate ecDNA from BFB? This question arises because Supplementary Table 1 reports the EGFR amplification in HCC827 is given as 'ecDNA+' by GCAP while AA reports it as "wild-type". More specifically, the source from which that comparison was drawn reported the sample as being a BFB.

4. An analysis of the kinds of focal amplifications GCAP predicts as ecDNA, but AA does not would be very useful (and vis-versa). For example, in batch 1 from Figure 2, there is one sample that is called ecDNA+ for GCAP but "wild-type" for AA, and in batch 2, two called as ecDNA+ for GCAP but "wild-type" for AA, as well as two AA ecDNA not called by GCAP. A deeper investigation into these disagreements would be helpful to understanding the limitations of both methods.

5. Non-genic ecDNA would be missed by GCAP's WES data. This limitation must be described somewhere in the manuscript. Better yet, a survey of the number of non-genic ecDNA predicted by AA that would not be detectable by GCAP in TCGA/PCAWG would help illustrate that point.

6. In the methods section (lines 696-700), the following is stated: "then TCGA cancer patients in GDC data portal with WES data and same patient identifiers were cross matched, of which 386 cancer patients (326 ecDNA positive and 60 ecDNA negative) were selected" - were the 60 ecDNA-negatives samples with no focal amplifications at all, or were they samples that also had some non-ecDNA focal amps? Again, it may be helpful to break ecDNA-negative into two classes - no focal amp and non-ecDNA focal amp.

7. Given that "total_cn" is by far the most important feature in the classifier, it stands to reason that non-ecDNA focal amps of high copy number may be prone to misclassification (false positive). As would low-copy number ecDNAs that have not yet undergone strong positive selective pressures (false negative). Both these limitations should be described in Discussion.

Minor issues:

- In Figure 2, the panel labelings are not correct. Panel e is given twice. Please fix and update the caption.

- Lines 87-88: "AmpliconArchitect... is limited by a sample size restriction, therefore, hinders our ability to study a broader range of samples beyond primary untreated tumor samples". This statement seems to be inaccurate. It is not clear what is meant by "sample size restriction", do the authors mean sample type restriction instead? AmpliconArchitect can be run on samples from primary tumor samples to metastases, cancer cell lines, and cancer models. Do the authors instead mean that because it is only compatible with paired-end WGS data it prevents them from analyzing their samples sequenced with WES? Please revise this description of AmpliconArchitect's limitations so that it is less misleading.

- Figure 4a-b, is the number next to the gene name a percentage of samples with that gene on ecDNA? A raw count? The caption states 'frequency' but the definition is not entirely clear. In cases where both blue and red exist on the same spot, it's hard to tell if both colors are there. Perhaps some additional transparency in the plotting to show both colors exist at that location may help.

- There is uncited overlapping work on mutational signatures and ecDNA status published in Hadi K, et al. "Distinct Classes of Complex Structural Variation Uncovered across Thousands of Cancer Genome Graphs." Cell, 2020 that should be considered.

- There are multiple grammatical and spelling errors (e.g. line 175 'litter' instead of 'lower') throughout introduction and other sections that must be corrected to improve readability and clarity.

Reviewer #3 (Remarks to the Author): clinical expertise in colorectal cancer

The authors gave a short description on how to detect ecDNA amplification from cancer genome sequencing data with computational toolkit AmpliconArchitect that has been developed to reconstruct fine circular structure of ecDNA from whole-genome sequencing (WGS) in silico, while Circle-Seq technique implements a sequencing library method for circular DNA specific enrichment to directly sequence possible circular DNA, followed by using software like Circle-Map to identify putative ecDNA junctions. The technique on how identification of ecDNA amplification from WES data could be described more in detail and especially the creation of the a focal amplification classifier, called gene-level circular amplicon prediction (GCAP), by utilizing an XGBOOST30 machine learning model that has been trained on gene-level copy number profiles, because this is the key of the method. Software such as AmpliconArchitect or Ampliconreconstructor, circle Finder and circmap are used to infer ecDNA structures from whole-genome sequencing data. These methods start by identifying regions of the genome with elevated copy number

and use those loci as a seed to construct a circular graph

WGS can be used to assemble ecDNA structures in silico but it remains a major computational challenge to distinguish chromosomal breakage–fusion–bridge structures from ecDNAs, coexisting homogeneously staining regions and coexisting homogeneously staining regions that have circularized, and ecDNAs in samples where some ecDNAs have reinserted into the genome.

More conservative , alternative methods should be also described to make the paper more complete. As well ecDNA-specific developments for ecDNA characterization e.g. circle-Seq, or the method for targeted profiling of ecDNA CRISPR-CATCH or RNA based methods like ecTag ould be mentioned. E.g. traditionally, cytogenetics methods have been used for ecDNA detection, including DAPI staining techniques and FISH.

The authors say: ".....We selected auPRC (area under precision-recall curve) as the primary evaluation metric on gene-level prediction due to the extreme class imbalance. As expected, the copy number, rather than other molecular profiles, exhibits moderate predictive ability (auPRC = 0.595) (Extended Data Fig. 2b). The data indicate that copy number is the most practical feature among the molecular signatures being investigated for ecDNA prediction..."

This formulation should be formulated more carefully, like The data may indicate that... normally ecDNAs are present at high copy number, which facilitates detection through sequencing-based

approaches, as the number of copies scales linearly with the number of derived sequencing reads,
.....supporting the sentence of the authors

Nature Communications Manuscript #NCOMMS-23-29647

Responses to Reviewers' comments

We would like to thank all the reviewers for their interest in our work and for their constructive comments and suggestions and believe their thoughtful input would lead to an improved version of our manuscript. Below we address the issues raised by each reviewer in a point-to-point way.

Reviewers' comments:

Reviewer #1:

In this manuscript, Wang and colleagues leverage a machine-learning based approach to infer ecDNA from whole exome sequencing (WES) data. They apply this tool to TCGA and PCAWG data sets and then focus on colorectal cancers, including data from 4 clinical trials that incorporate PD-1 immunotherapy. They show that ecDNA is associated with worse prognosis in patients, and that it is an independent risk factor in colorectal cancers, and also reveal that patients whose tumors have ecDNA have derive less benefit from anti-PD-1 therapy.

The authors have done a great deal of impressive work and many aspects of the paper are potentially very important and of high interest, including developing a way to infer ecDNA status and its contents from WES data; and showing its adverse impact on patients with colorectal cancers, including in response to anti-PD1 therapy.

General response: We thank the reviewer for taking the time to review our manuscript. We appreciate the reviewer's recognition of the significance of our study in advancing the field of cancer research. We would like to address the comments and suggestions provided by the reviewer to improve the manuscript's suitability for publication. We believe that incorporating these changes will enhance the quality of our work. Thank you once again for your valuable input, which will undoubtedly help us to strengthen our research findings.

There are however, significant concerns that need to be addressed, as follows:

1) Validation of the machine learning approach to make ecDNA-calls: Overall, the numbers make great sense and the comparison with Amplicon Architect data is compelling. However, the direct validation of the method is very limited. It seems essentially to boil down to FISH on 2 known cancer cell lines and 2 colorectal cancer clinical samples. While this is nice, it is not nearly adequate for validating this as an independent tool for ecDNA calling.

Response: We appreciate the reviewer's thoughtful evaluation of our validation process. In response to the concern, we have made considerable efforts to bolster the validation of our machine learning approach for ecDNA amplification detection. We have expanded our validation strategy in several key ways:

1. Clinical FFPE Samples: We performed FISH analysis on an additional 12 FFPE samples obtained from 11 colorectal cancer patients, all of whom were previously identified as ecDNA+ for cargo genes *MYC* or *ERBB2* by GCAP. Through manual inspection, we consistently observed diffuse or clustered gene amplification signals (Supplementary file 2). This extended validation reinforces the reliability of GCAP in clinical samples.
2. New Gastric Cancer Cell Line (SNU216): We conducted whole-genome sequencing (WGS) on the newly included gastric cancer cell line, SNU216. Our analysis results, both based on AmpliconArchitect (AA) and GCAP, consistently identified it as ecDNA+. We have integrated this new data into our Supplementary Table 1 and Fig. 2f.
3. Circle-Seq Analysis: We conducted Circle-Seq analysis on 11 cancer cell lines predicted to have ecDNA amplifications by GCAP (Supplementary Table 1). As anticipated, all these cell lines successfully met the stringent Circle-Map filtering criteria, with mitochondrial DNA sequences serving as positive controls (Supplementary Table 2). When we mapped the GCAP-predicted ecDNA cargo genes to the corresponding genomic regions with Circle-Seq sequencing depth data, we observed that these genes either located at local peaks (including oncogenes *MYC*, *FGFR2*, *CTTN*, *ERBB2*, *JUP*, *CCNE1*, *MET*, etc.) or situated within regions with sufficient coverage (Extended Data Fig. 4). This direct evidence further substantiates GCAP's validity at both the sample and gene levels.

Furthermore, it's important to note that the results presented in Fig. 2e, involving the WGS and WES data analysis of a gastric cancer tissue sample and PDX sample, also serve as direct

validations of the GCAP method.

In summary, our expanded validation efforts, encompassing clinical samples, additional cell lines, and Circle-Seq data, collectively demonstrate the robustness and practical utility of GCAP for ecDNA amplification detection.

2) Description of the machine learning tool is very limited: The figure is nice, but in fact, it is very hard to figure out what they actually did. A more detailed treatment, even in supplement, would be very important.

Response: We appreciate the reviewer's valuable feedback, which has significantly enhanced the integrity of our study. In response to the comments, we have added Supplementary file 1 to our revised submission. This supplementary document provides comprehensive information on our data collection and preprocessing methods, details our modeling process, outlines the implementation framework utilizing R packages, and offers a concise guide for utilizing the developed R packages.

3) Application to colorectal subtypes: The authors have previously published work on colorectal subtyping. However, these distinctions are not easily intuitive. A reader of this paper will now easily know what LR or HR mean in this context. The treatment feels cursory.

Response: We appreciate the reviewer's constructive feedback regarding the clarity of our colorectal subtypes terminology. To address this concern, we have revised the introduction to the SYSUCC CRC subtypes and figure legend of Fig. 5, providing a more detailed explanation of the abbreviations and their meanings for better reader comprehension. Specifically, LR and HR represent 'low risk' and 'high risk', respectively. Here is the updated introduction:

'Previously, we established a Sun Yat-sen University Cancer Center (SYSUCC) CRC subtypes system, which divided the CRC into subtype HM (**H**yper**M**utated), subtype GS (**G**enome **S**table), subtype CIN-LR (**C**hromosomal **I**Nstability with **L**ow survival **R**isk) and subtype CIN-HR (**C**hromosomal **I**Nstability with **H**igh survival **R**isk). This classification was mainly based on the somatic single-nucleotide variations (SNVs) and copy number variations (CNVs). For ecDNA, due

to the limited sample size of CRC patients in the study by Kim *et al.* study, uncertainties remain regarding the prevalence of ecDNA amplification in CRC, the correlation between ecDNA amplification and clinical characteristics of CRC, and whether focal amplification typing could provide insights into CRC heterogeneity. To address these gaps, we sought to leverage the GCAP tool to analyze a comprehensive dataset of 1,015 patients from the SYSUCC CRC cohort, which were whole-exome sequenced as part of the Changkang (Healthy Bowel) Project.'

4) Mutational signatures component: this part is very superficial. More details or illustration is greatly needed. At present, it detracts rather than adds. It is clearly an important topic, but not well suited to cursory treatment.

Response: We appreciate the reviewer's feedback and understand the concern regarding the depth of the mutational signatures component in our manuscript. Upon careful consideration, we have taken steps to address this issue. We acknowledge that the initial treatment of this topic may have been too superficial for a comprehensive understanding. In response, we have removed the paragraph dedicated to mutational signatures in the revised manuscript.

However, we recognize the importance of this topic and have retained the relevant result data in Supplementary Table 4 and 5 for readers who may be interested in exploring the mutational signatures in more detail. To better align with the main focus of our study, we have shifted our emphasis to the analysis of ecDNA+ associated APOBEC mutational signatures and copy number signature No.8 (CN8) (Fig. 5f, g). These analyses independently validate our findings from previous pan-cancer analyses, enhancing the robustness of our study.

Furthermore, we have conducted a comparative analysis of gene mutations between cancer patients with circular amplifications and those without. Notably, we found a significant association between *TP53* mutations and ecDNA amplifications (Extended Data Fig. 5 and Fig. 5e). This observation is consistent with prior research indicating that *TP53* alterations play a role in enabling the development of ecDNA during the cancerous transformation of Barrett's esophagus.

We believe that these adjustments improve the overall quality and relevance of our manuscript while addressing the reviewer's concerns about the mutational signatures component. We hope these changes align more closely with the objectives of our study and provide valuable insights for

our readers.

5) Major concern regarding interpretation of combination therapy data related to Extended Figure 11: They attempt to show that addition of immunotherapy to chemotherapy adds benefit preferentially to patients whose tumors have ecDNA. However, this reviewer believes that this is an incorrect conclusion. Analysis of the numbers reveals that the number of patients with ecDNA in the trial is roughly double that of patients with non-circular amplification or no amplification. The curves look very similar but only the ecDNA containing tumor curve reaches significance, likely because it has double the numbers. This has truly important impact on the conclusion. Doesn't that suggest that addition of immunotherapy can help in these esophageal cancer patients in all 3 examples, even if ecDNA-containing tumors are less response to IO monotherapy? Positing enhanced benefit in ecDNA-containing tumors by adding IO therapy may be a misleading conclusion.

Response: We appreciate the reviewer's valuable input. The statement regarding the "addition of immunotherapy to chemotherapy adds benefit preferentially to patients whose tumors have ecDNA" appears to have generated some misunderstandings. We understand that the reviewer intended to convey that not only patients with ecDNA amplification can benefit but that there is also an observed trend of improved survival in other groups. We acknowledge that our previous presentation and discussions may have disproportionately emphasized the outcomes of patients with ecDNA amplification. In response, we have undertaken significant revisions in the manuscript to provide a more comprehensive description of our analyzed results. Additionally, we have conducted a more in-depth analysis at the cytoband level to further validate our findings. Our data unquestionably demonstrate that, at least within the JUPITER-06 cohort, there is indeed an enhanced benefit in ecDNA-containing tumors when immunotherapy is added on the basis of chemotherapy.

Here is our updated result part: "In our extended focal amplification typing analysis, we found that the circular subtype exhibits a higher response rate (complete response + partial response) and a lower rate of progressive disease in both treatment groups (Extended Data Fig. 13a). When examining different focal amplification subtypes, we observed that patients treated with

chemo+immunotherapy showed an overall survival improvement compared to those treated with chemo+placebo across all three subtypes. This improvement was statistically significant in the circular subtype (hazard ratio = 0.52, $P = 0.0054$, Extended Data Fig. 13b). Furthermore, a comparison of Kaplan-Meier curves across all subgroups revealed that patients with the circular subtype treated with chemo+immunotherapy achieved the best overall survival. When excluding this subgroup from the analysis, no significant differences were observed in the remaining subgroups (Extended Data Fig. 13c). Notably, our cytoband-level analysis identified 11q13 (containing oncogenes such as *CCND1* and *CTTN*) as the dominant focal amplification in the cohort (Extended Data Fig. 14a, b). Survival analysis specific to this cytoband showed that patients treated with chemo+immunotherapy experienced a significant overall survival improvement compared to those treated with chemo+placebo in the circular and nofocal subtypes, but not in the noncircular subtype (Extended Data Fig. 14c). Collectively, these results suggest that ecDNA amplifications, particularly those linked to 11q13, exhibit greater sensitivity to chemo+immunotherapy compared to chemotherapy alone in advanced ESCC.”

In the discussion section, we have restructured our focus away from the previous discussion on immune checkpoint inhibitor combination therapy. Instead, we have directed attention towards highlighting the distinct responses observed in the context of anti-PD1 monotherapy and combination therapy, particularly within the 11q13 cytoband region with ecDNA amplification. Furthermore, we have expanded upon the limitations of our analysis to offer a more comprehensive and detailed explanation.

Here is our updated discussion part: “Interestingly, we also observed a distinct trend among advanced esophageal squamous cell carcinoma patients with ecDNA amplification. When subjected to chemo+immunotherapy, these patients demonstrated a significant improvement in overall survival compared to those receiving chemo+placebo treatment. Notably, the presence of ecDNA amplifications on 11q13 was found to be predictive of poor outcomes in the SYSUCC NPC cohort when treated with anti-PD1 monotherapy (Extended Data Fig. 12d), whereas it predicted favorable survival outcomes in the JUPITER-06 cohort undergoing anti-PD1 plus chemotherapy treatment (Extended Data Fig. 14c). This transformation from nonresponse to response may be attributed to the synergistic effects of chemotherapy-induced immunogenic cell death (ICD), therapy-induced neoantigen release, and the modulation of the immunosuppressive tumor microenvironment

through anti-PD1 immunotherapy. However, it is crucial to exercise caution in interpreting these findings. We must emphasize that the JUPITER-06 cohort represents the sole clinical trial encompassing gastrointestinal cancer patients subjected to chemo+immunotherapy combination treatment with available WES data. Nonetheless, it is essential to acknowledge the potential limitations associated with the uneven sample sizes across different focal amplification subtypes within this cohort. Taken together, it is imperative to consider the specific treatment strategy employed when evaluating the biomarker value of ecDNA amplification for patient outcomes. To gain a comprehensive understanding of the broader implications of this biomarker, additional clinical trials and data analyses involving various cancer types, particularly those characterized by immune-hot tumor microenvironments (e.g., melanoma, non-small cell lung cancer), are warranted. These endeavors will facilitate a more comprehensive assessment of the utility of ecDNA amplification as a predictive marker in diverse clinical contexts.”

We believe that these updates enhance the clarity and depth of our analysis and discussion regarding interpretation of combination therapy data, effectively addressing the reviewer's concerns. We appreciate the reviewer's valuable input in enhancing the quality of our manuscript.

Reviewer #2:

Wang, et al. "Machine learning-based extrachromosomal DNA identification in large-scale cohorts reveals its clinical implications in cancer" presents a novel framework for the identification of ecDNA based on whole-exome sequencing, copy number calling and prior analysis of whole genome sequencing to predict the gene cargo of ecDNA. Wang et al. analyze the relationship of ecDNA with gene cargo, survival, mutational signature, and anti-PD1 immunotherapy. The paper reinforces existing knowledge of ecDNA gene contents, transcriptional dynamics and association with survival, and also presents an important step forward in our understanding of how ecDNA presence affects immunotherapy.

General response: We sincerely thank the reviewer for their diligent review of our manuscript. We are grateful for the reviewer's concise summary and their recognition of the importance of our study in advancing the field of ecDNA, particularly in the context of cancer immunotherapy. We are committed to addressing the reviewer's insightful comments and suggestions to enhance the

manuscript's suitability for publication. Your valuable input is greatly appreciated, and we are confident that it will significantly contribute to improving the quality and impact of our work.

Major issues:

1. The use of the word "wild-type" is misleading when describing non-ecDNA calls (for example in Figure 2, Supplementary Table 1, Figure 6). Rather it seems they are simply ecDNA-. Only if the copy number is measured to be normal is the label "wild-type" appropriate. Many are likely non-ecDNA focal amps, and that is certainly not a "wild-type" state.

Response: We appreciate the reviewer's valuable input regarding the terminology used in our study. We acknowledge that the use of 'wild-type' to describe non-ecDNA calls may be misleading. It is essential to accurately represent the different states. Upon careful consideration and taking into account the reviewer's suggestion, we have revised our terminology to address this concern appropriately. Correspondingly, we updated the word "wild-type" to "ecDNA-" in Fig. 1, Fig. 2, Fig. 6, Supplementary Table 1, and texts.

2. In the analysis of ecDNA status and immunotherapy response (Figure 6), the analysis is performed as ecDNA+ vs. "wild-type". In reality, there are two hidden classes in the "wild-type" category; non-ecDNA focal amps, and no focal amp at all. Is there any difference in response between ecDNA+ and non-ecDNA focal amps? Specifically, does it matter that oncogene amplification is carried on ecDNA when gauging anti-PD1 response or is any mechanism of amplification sufficient to hinder response? Does it matter which genes are amplified in determining the response?

Response: We appreciate the reviewer's important comments and have made significant updates to address them.

In response to the first question, we conducted a survival curve comparison among different focal copy number amplification (fCNA) types, including circular, noncircular, and nofocal, as shown in Response Figure 1. Our analysis revealed that patients with circular amplification had a worse

survival outcome than patients with noncircular amplification in both the SYSUCC NPC cohort (Hazard ratio = 1.21, $P = 0.581$) and the SYSUCC AGC cohort (Hazard ratio = 3.77, $P = 0.004$). These statistical results have been incorporated into Fig. 6c and 6d in the revised submission. We also observed that nofocal patients had a poor outcome in the SYSUCC AGC cohort. However, this could potentially be influenced by covariates such as prior treatment lines, which might divert our focus from presenting the data related to ecDNA amplifications. Therefore, we have decided to maintain the comparison between circular+ and circular- in Fig. 6a and 6b, while Fig. 6c and 6d provide a comprehensive analysis and comparison between fCNA subtypes and other variables through forest plots generated from multivariable Cox analysis.

Regarding the second question, we considered the low gene ratio with focal amplifications and relatively high gene heterogeneity across different cancer cohorts (Response Figure 2), even in the same cancer type – advanced gastric cancer. Therefore, analyzing and presenting gene-level data in the manuscript may not be suitable, as we initially considered. However, we conducted an analysis at the cytoband-level and identified two specific cytobands, 8q24 (containing oncogene *MYC*) in the SYSUCC AGC cohort and 11q13 (containing oncogenes *CCND1*, *CTTN*, etc.), where patients with ecDNA amplifications in these regions exhibited significantly worse survival outcomes compared to patients without ecDNA amplifications in these regions (Extended Data Fig. 12). While the sample size of patients with ecDNA amplifications in these regions is relatively small, we did not observe a significant difference in survival outcomes for other top amplified cytobands. We have incorporated these findings into both the Results section and the Discussion section of the revised manuscript, providing a careful description and discussion of these observations.

We believe that these updates enhance the clarity and depth of our analysis regarding immunotherapy response and ecDNA status, effectively addressing the reviewer's concerns.

Response Figure 1 to Reviewer 2. Gene copy number mean vs. gene count in three anti-PD1 cancer cohorts.

Genes with high copy number or multiple occurrences are labeled.

Response Figure 2 to Reviewer 2. Gene copy number mean vs. gene count in three anti-PD1 cancer cohorts.

Genes with high copy number or multiple occurrences are labeled. (Note: zoom in for better resolution)

3. Discriminating breakage-fusion-bridge focal amps (BFBs) from ecDNAs is a computationally challenging task due to the high genomic copy numbers found in both events. Can the authors show that their method is able to reliably discriminate ecDNA from BFB? This question arises because Supplementary Table 1 reports the EGFR amplification in HCC827 is given as 'ecDNA+' by GCAP while AA reports it as "wild-type". More specifically, the source from which that comparison was drawn reported the sample as being a BFB.

Response: We deeply appreciate the reviewer's perceptive inquiry regarding the intricate task of discriminating breakage-fusion-bridge focal amplifications (BFBs) from extrachromosomal circular DNA (ecDNAs). As highlighted by another reviewer, distinguishing between these structures, especially in scenarios with high genomic copy numbers and intricate genomic events, remains a formidable computational challenge in the field of cancer genomics. Recent research led by pioneers in the ecDNA field, Paul S. Mischel and Vineet Bafna, has shed light on the complexities associated with this distinction. Their study, which leveraged TCGA transcriptome data, underscores the limitations stemming from short-read sequencing. In such cases, AmpliconArchitect (AA) may erroneously classify certain ecDNA+ structures as complex non-cyclic when breakpoints are missed. Moreover, the potential for BFB cycles to give rise to ecDNA adds further complexity to discriminating between these modes of amplification. To mitigate the risk of

false-negative ecDNA classifications in the ecDNA- dataset, their study adopted a strategy of considering samples with only linear or no amplification as ecDNA- (similar to our preprocessing for constructing the modeling data), thereby excluding complex non-cyclic and BFB+ samples from the analysis (Lin, Miin S., *et al.* "Transcriptional immune suppression and upregulation of double-stranded DNA damage and repair repertoires in ecDNA-containing tumors." *bioRxiv* (2023): 2023-04). It is noteworthy that Paul S. Mischel and Vineet Bafna are also responsible for the creation of the AA software employed in our study (Deshpande, Viraj, *et al.* "Exploring the landscape of focal amplifications in cancer using AmpliconArchitect." *Nature communications* 10.1 (2019): 392).

Acknowledging these complexities and the inherent computational challenges, we concede that distinguishing between ecDNA and BFBs poses a formidable task. This complexity arises from multiple factors, including GCAP's predominant reliance on copy number features, which may result in BFBs being misclassified as ecDNAs. Additionally, the inherent ambiguity in AA's classification of ecDNA and BFB can further compound GCAP's performance in this context. From a biological perspective, BFBs may serve as mechanisms inducing ecDNA formation, and their coexistence and mutual transformation have been observed. As such, we believe that rigorously enforcing a strict computational demarcation between these two entities may not yield substantial analytical benefits for cancer cohorts. When specific research objectives necessitate their differentiation, a nuanced approach that combines computational methodologies and experimental techniques can offer valuable insights into the underlying biological mechanisms, especially in cell line models.

Furthermore, in response to the reviewer's specific example concerning *EGFR* amplification in HCC827, we conducted an examination of the AA detection results (Response Table 1), employing default parameters. Notably, these AA results did not yield any BFB detections within HCC827. Conversely, our AA analysis identified a "Complex non-cyclic" event within the genomic region containing the *EGFR* gene. In contrast, the AR software, relying on whole-genome sequencing (WGS) and optical mapping data, corroborated the presence of a BFB structure encompassing the *EGFR* gene in HCC827 (Luebeck, Jens, *et al.* "AmpliconReconstructor integrates NGS and optical mapping to resolve the complex structures of focal amplifications." *Nature communications* 11.1 (2020): 4374). This discrepancy underscores the limitations inherent in short-read sequencing-based ecDNA analysis, which can be influenced by both sequencing methodologies and data quality.

Finally, we conducted an additional analysis using PCAWG data to evaluate GCAP's capacity to

discriminate between BFBs and ecDNAs as detected by AA (Response Table 2). Within a dataset comprising 197 matched and analyzed patient samples, 121 samples were classified as ecDNA+ by AA, with 19% of them exhibiting both circular and BFB amplicons. Utilizing AA's results as a reference, we conducted a straightforward analysis to assess GCAP's discriminative performance, yielding the following metrics: Accuracy: 0.548, Specificity: 0.724, Sensitivity: 0.438. These metrics indicate that GCAP possesses limited discriminatory capability when distinguishing between ecDNAs and BFBs.

In summary, we extend our sincere gratitude to the reviewer for their insightful inquiry, and we acknowledge the intricate computational and biological challenges associated with distinguishing ecDNAs from BFBs. We have undertaken a comprehensive exploration that addresses all of the issues raised.

id	AmpliconID	amplicon_decomposition_classes	ecDNA+	BFB+	ecDNA_amplicons	No amp/Invalid	Linear amplification	Trivial cycle	Complex non-cyclic	Intervals	OncogenesAmplified
HCC827	10	Linear amplification	None detected	None detected	0	0.129011472	0.870988528	0	0	chr12:622452-66-62495266	,
HCC827	11	Linear amplification	None detected	None detected	0	0.138988987	0.652154069	0	0.208856944	chr12:643952-68-65030268	,
HCC827	12	Linear amplification	None detected	None detected	0	0	1	0	0	chr12:656252-68-65965268	HMGA2,
HCC827	13	Linear amplification	None detected	None detected	0	0.294149383	0.705850617	0	0	chr18:357954-60-36430440	,
HCC827	14	Linear amplification	None detected	None detected	0	0.369552897	0.630447103	0	0	chr18:385253-26-38825321	,
HCC827	15	Linear amplification	None detected	None detected	0	0	1	0	0	chr21:150909-29-18751118	,
HCC827	16	No amp/Invalid	None detected	None detected	0	1	0	0	0	chrX:5387466-5-54704827	,
HCC827	17	No amp/Invalid	None detected	None detected	0	1	0	0	0	chrX:5793545-7-58100000	,
HCC827	1	Linear amplification	None detected	None detected	0	0	1	0	0	chr6:1139478-2-11884775	,
HCC827	2	Complex non-cyclic	None detected	None detected	0	0	0.463604094	0	0.536395906	chr7:5326468-1-5634667	EGFR,
HCC827	3	Linear amplification	None detected	None detected	0	0.03090537	0.96909463	0	0	chr7:5639465-9-58100000	,
HCC827	4	Complex non-cyclic	None detected	None detected	0	0.193262062	0.334013126	0	0.472724812	chr8:6577824-7-67638329,chr8:124753750-136023797	NDRG1,MYC,
HCC827	5	Linear amplification	None detected	None detected	0	0.008425499	0.757691048	0	0.233883454	chr8:6952343-2-70413465,chr8:73048605-75193704	NCOA2,
HCC827	6	Linear amplification	None detected	None detected	0	0	1	0	0	chr8:7972393-4-81904035	HEY1,
HCC827	7	Linear amplification	None detected	None detected	0	0	0.159032702	0	0.840967298	chr8:1017240-62-103249049,chr8:123298850-123638848	UBR5,
HCC827	8	Linear amplification	None detected	None detected	0	0	1	0	0	chr12:542402-62-54590262	,
HCC827	9	Linear amplification	None detected	None detected	0	0.121314316	0.878685684	0	0	chr12:574702-64-57855264	DDIT3,CDK4,

Response Table 1 to Reviewer 2. AA detection results for the cancer cell line HCC827. The row corresponding to *EGFR* is highlighted for emphasis.

	GCAP – ecDNA-	GCAP – ecDNA+
AA – BFB	55	21
AA - ecDNA	68	53

Response Table 2 to Reviewer 2. Evaluation of GCAP's capacity to discriminate between BFBs and ecDNAs as detected by AA.

4. An analysis of the kinds of focal amplifications GCAP predicts as ecDNA, but AA does not would be very useful (and vis-versa). For example, in batch 1 from Figure 2, there is one sample that is called ecDNA+ for GCAP but "wild-type" for AA, and in batch 2, two called as ecDNA+ for GCAP but "wild-type" for AA, as well as two AA ecDNA not called by GCAP. A deeper investigation into these disagreements would be helpful to understanding the limitations of both methods.

Response: We thank the reviewer for the valuable suggestion. We have conducted an in-depth exploration of the discrepancies between GCAP and AA, especially in cancer cell lines with inconsistent results (Response Table 3), and we have identified several factors that may contribute to these inconsistencies:

1. Accuracy of Copy Number Detection: Copy number information is pivotal for both AA and GCAP methods. AA leverages copy number detection results to identify seed intervals and subsequently pinpoint breakpoints supporting ecDNA, facilitating the reconstruction of ecDNA structures. Copy number is also a vital feature in GCAP's prediction model. Although the utilization of whole-exome sequencing (WES) data for copy number variation detection is widely endorsed in the academic community, it's important to note that the precision of copy number detection across diverse genomic regions in various samples can be influenced by factors such as sequencing data quality and computational methodologies. While we incorporated the widely accepted ASCAT method in our standard pipeline for copy number detection from WES data, we acknowledge that achieving 100% accuracy in detecting copy number profiles for all genes is challenging. Fortunately, this limitation did not compromise the overall integrity of our study, as it allowed us to attain a robust classification that ensured the generation of biologically meaningful results during our analyses.

2. Limitations of GCAP: GCAP software implementation has limitations. On one hand, the current GCAP software cannot be applied to non-coding genomic elements, as mentioned by the reviewer in point 5. On the other hand, as highlighted by the reviewer in point 7, the importance of the "total_cn" feature in the GCAP classifier suggests that non-ecDNA focal amplifications with

high copy numbers may be prone to misclassification (false positives, e.g., HCC827 and SH10TC), as well as low-copy-number ecDNAs that have not yet undergone strong positive selective pressures (false negatives, e.g., FU97 and KYSE180). We acknowledge the reviewer's assessment, and it is an area we plan to explore for improvement in the future.

3. Limitations of AA: Given the inherent uncertainty in ecDNA detection (e.g., AA may classify some ecDNA+ structures as complex non-cyclic when breakpoints are missed), AA may not yield definitive classification results for all cases (e.g., in the case of cell line NCIN87, Circle-Seq identifies it as ecDNA+, in contrast to the AmpliconArchitect (AA) prediction, which suggests it to be ecDNA-, Supplementary Table 1 and Extended Data Fig. 4). Utilizing pseudo-labels generated from AA classifications for machine learning modeling may result in a reduction in performance. However, achieving both an absolute reference (ground truth) and exceptionally high detection performance simultaneously poses a current challenge. Existing ecDNA detection methods struggle to definitively identify all true positive and true negatives. Nonetheless, our primary goal in this study is to attain robust ecDNA amplification classification for meaningful biological analysis. Our data illustrate that classifiers constructed using AA-generated pseudo-labels remain a viable approach, consistently producing valuable results.

We have incorporated a comprehensive discussion of the limitations of GCAP in our revised manuscript. We believe that these additional insights enhance our understanding of the limitations and complexities of ecDNA amplification detection and classification methods.

Cell line	AA	GCAP
HCC827	ecDNA-	ecDNA+
FU97	ecDNA+	ecDNA-
KYSE180	ecDNA+	ecDNA-
NCIN87	ecDNA- (BFB)	ecDNA+
SH10TC	ecDNA- (BFB)	ecDNA+

Response Table 3 to Reviewer 2. Cancer cell lines with inconsistent ecDNA status detected by AA and GCAP.

5. Non-genic ecDNA would be missed by GCAP's WES data. This limitation must be described

somewhere in the manuscript. Better yet, a survey of the number of non-genic ecDNA predicted by AA that would not be detectable by GCAP in TCGA/PCAWG would help illustrate that point.

Response: We are grateful for the reviewer's insightful suggestion and have taken steps to address this limitation in our manuscript. While our primary focus has been on genic regions, we acknowledge that non-genic ecDNA elements exist, and they could be missed by GCAP's WES data analysis.

In response to the reviewer's suggestion, we conducted an abundance analysis of non-genic features in both TCGA/PCAWG data. Our analysis, presented in Extended Data Figure 15, reveals that approximately two-thirds of ecDNA originates from non-genic regions. This significant proportion underscores the importance of considering non-genic ecDNA elements in comprehensive ecDNA studies.

Furthermore, we have incorporated a discussion of this limitation in our revised manuscript, which given as "Additionally, it's essential to highlight that the current version of GCAP does not encompass noncoding regions within its scope, even though noncoding elements constitute approximately two-thirds of ecDNA amplifications (Extended Data Fig. 15)", emphasizing the significance of non-genic ecDNA and the potential challenges in detecting them with our current implementation.

We believe that these additions enhance the clarity and completeness of our manuscript and address the reviewer's valuable input.

6. In the methods section (lines 696-700), the following is stated: "then TCGA cancer patients in GDC data portal with WES data and same patient identifiers were cross matched, of which 386 cancer patients (326 ecDNA positive and 60 ecDNA negative) were selected" - were the 60 ecDNA-negatives samples with no focal amplifications at all, or were they samples that also had some non-ecDNA focal amps? Again, it may be helpful to break ecDNA-negative into two classes - no focal amp and non-ecDNA focal amp.

Response: We appreciate the reviewer's insightful question regarding the classification of ecDNA-negative samples in our methods description. To provide clarity and transparency, we have added

a Supplementary file 1, which provides a comprehensive overview of our modeling process.

Regarding the categorization of ecDNA-negative samples, we made a distinction based on the presence or absence of focal amplifications. Given several considerations, including the prevalence of ecDNA-negative tumors, the potential complexity of categorizing ecDNA structures, and the resource-intensive nature of gene-level data modeling, we took the following approach:

1. We retained a total of 326 matched ecDNA+ tumors.
2. Additionally, we randomly selected 30 ecDNA- tumors that exhibited 'Linear' amplification, representing non-ecDNA focal amplification.
3. We also randomly selected 30 ecDNA- tumors labeled as 'No-fSCNA,' representing samples with no focal amplification.

This approach allows us to differentiate between ecDNA-negative samples with non-ecDNA focal amplifications and those without any focal amplifications (similar to the study by Lin, Miin S., *et al.* "Transcriptional immune suppression and upregulation of double-stranded DNA damage and repair repertoires in ecDNA-containing tumors." *bioRxiv* (2023): 2023-04). We believe that this classification enhances the precision and accuracy of our analysis while addressing the reviewer's concern.

7. Given that "total_cn" is by far the most important feature in the classifier, it stands to reason that non-ecDNA focal amps of high copy number may be prone to misclassification (false positive). As would low-copy number ecDNAs that have not yet undergone strong positive selective pressures (false negative). Both these limitations should be described in Discussion.

Response: We thank the reviewer for their insightful observation regarding the classifier's reliance on 'total_cn' as a crucial feature. It is indeed reasonable to consider the potential implications of this feature's importance on the classification process.

To address this concern, we have added a discussion of these limitations in our manuscript. Specifically, we acknowledge that non-ecDNA focal amplifications with high copy numbers may be more susceptible to misclassification, leading to potential false positives. Conversely, low-copy number ecDNAs that have not yet undergone strong positive selective pressures may be prone to false negatives.

By highlighting the limitation in the Discussion section, which given as "This design choice has the potential to introduce elevated levels of false positives and false negatives, particularly given the reliance on copy number as a pivotal predictive feature", we aim to provide a comprehensive perspective on the classifier's performance and its potential implications in real-world scenarios. We appreciate the reviewer's valuable input in enhancing the quality of our manuscript.

Minor issues:

- In Figure 2, the panel labelings are not correct. Panel e is given twice. Please fix and update the caption.

Response: We thank the reviewer for pointing out this error. We have fixed it and updated the caption in the revised submission.

- Lines 87-88: "AmpliconArchitect... is limited by a sample size restriction, therefore, hinders our ability to study a broader range of samples beyond primary untreated tumor samples". This statement seems to be inaccurate. It is not clear what is meant by "sample size restriction", do the authors mean sample type restriction instead? AmpliconArchitect can be run on samples from primary tumor samples to metastases, cancer cell lines, and cancer models. Do the authors instead mean that because it is only compatible with paired-end WGS data it prevents them from analyzing their samples sequenced with WES? Please revise this description of AmpliconArchitect's limitations so that it is less misleading.

Response: We appreciate the reviewer's critical comment and the correct understanding of our content. To reduce confusion, we have revised the sentence as follows: "AmpliconArchitect is designed solely for WGS data, which limits our ability to study a broader range of tumor samples from clinical cohorts that are typically sequenced by whole-exome sequencing (WES)". This revision provides a clearer explanation of the limitations of AmpliconArchitect without introducing any ambiguity.

- Figure 4a-b, is the number next to the gene name a percentage of samples with that gene on ecDNA? A raw count? The caption states 'frequency' but the definition is not entirely clear. In cases where both blue and red exist on the same spot, it's hard to tell if both colors are there. Perhaps some additional transparency in the plotting to show both colors exist at that location may help.

Response: We appreciate the reviewer's valuable suggestions. To address the query regarding Figure 4a-b, we have modified the caption to specify that the number next to the gene name represents 'the count' of samples with that gene on ecDNA rather than 'frequency'.

Regarding the issue of differentiating between blue and red in cases where both colors exist at the same location, we would like to clarify that the genome-level heatmap was generated using a plot function from our `gcaputils` package, utilizing the R package `ComplexHeatmap` and following the guidelines outlined in the 'Genome-level heatmap' section of the `ComplexHeatmap Complete Reference` (<https://jokergoo.github.io/ComplexHeatmap-reference/book/genome-level-heatmap.html>).

In this context, a function called `average_in_window()` was created by the author of `ComplexHeatmap` to visualize genome-scale signals as a heatmap by calculating the average signals within 1MB windows (i.e., the same spot). When a spot exhibits both circular and noncircular signals, one signal may appear to be shadowed by the other. Consequently, a region can only be colored in either red or blue, or grey (for none), but not simultaneously in both red and blue. To differentiate between circular and noncircular signals in such cases, readers can refer to the two bar plots on the right side of the heatmap. To provide further clarity and reduce any potential confusion, we have updated the figure caption accordingly.

- There is uncited overlapping work on mutational signatures and ecDNA status published in Hadi K, et al. "Distinct Classes of Complex Structural Variation Uncovered across Thousands of Cancer Genome Graphs." *Cell*, 2020 that should be considered.

Response: We appreciate the reviewer's input and have included the relevant citation to the work by Hadi K *et al.* ('Distinct Classes of Complex Structural Variation Uncovered across Thousands of Cancer Genome Graphs,' *Cell*, 2020) in our manuscript to properly acknowledge and reference the

overlapping work on mutational signatures and ecDNA status.

- There are multiple grammatical and spelling errors (e.g. line 175 'litter' instead of 'lower') throughout introduction and other sections that must be corrected to improve readability and clarity.

Response: We appreciate the reviewer for bringing this to our attention. We have addressed the error (changing 'litter' to 'little') and conducted a thorough review of the entire manuscript with multiple individuals to identify and rectify any potential grammar and spelling issues. These revisions have been made to enhance the readability and clarity of the manuscript.

Reviewer #3:

The authors gave a short description on how to detect ecDNA amplification from cancer genome sequencing data with computational toolkit AmpliconArchitect that has been developed to reconstruct fine circular structure of ecDNA from whole-genome sequencing (WGS) in silico, while Circle-Seq technique implements a sequencing library method for circular DNA specific enrichment to directly sequence possible circular DNA, followed by using software like Circle-Map to identify putative ecDNA junctions. The technique on how identification of ecDNA amplification from WES data could be described more in detail and especially the creation of the a focal amplification classifier, called gene-level circular amplicon prediction (GCAP), by utilizing an XGBOOST30 machine learning model that has been trained on gene-level copy number profiles, because this is the key of the method. Software such as AmpliconArchitect or Ampliconreconstructor, circle Finder and circmap are used to infer ecDNA structures from whole-genome sequencing data. These methods start by identifying regions of the genome with elevated copy number and use those loci as a seed to construct a circular graph

General response: We sincerely appreciate the thorough review conducted by the reviewer. We are committed to addressing the reviewer's valuable comments and suggestions to improve the manuscript's overall clarity and comprehensiveness.

In response to the reviewer's suggestion to provide a more detailed description of the method for identifying ecDNA amplification from whole-exome sequencing (WES) data, we have added a Supplementary file 1 to our revised submission. This supplementary document provides comprehensive information on our data collection and preprocessing methods, details our modeling process, outlines the implementation framework utilizing R packages, and offers a concise guide for utilizing the developed R packages. We are confident that these enhancements will contribute to the manuscript's quality and impact.

We express our gratitude once again for the reviewer's insightful feedback, which will undoubtedly strengthen the robustness of our research findings.

WGS can be used to assemble ecDNA structures in silico but it remains a major computational challenge to distinguish chromosomal breakage–fusion–bridge structures from ecDNAs, coexisting homogeneously staining regions and coexisting homogeneously staining regions that have circularized, and ecDNAs in samples where some ecDNAs have reinserted into the genome.

More conservative , alternative methods should be also described to make the paper more complete.

As well ecDNA-specific developments for ecDNA characterization e.g. circle-Seq, or the method for targeted profiling of ecDNA CRISPR-CATCH or RNA based methods like ecTag could be mentioned. E.g. traditionally, cytogenetics methods have been used for ecDNA detection, including DAPI staining techniques and FISH.

Response: We appreciate the reviewer's valuable suggestion to enhance the comprehensiveness of our study. To address this concern, we have added a section to the introduction that describes various alternative methods for ecDNA characterization:

“Methods for ecDNA characterization are crucial for both fundamental and applied research on ecDNA. Traditional cytogenetic techniques, such as 4',6-diamidino-2-phenylindole (DAPI) staining and fluorescence in situ hybridization (FISH), have been employed to detect and quantify ecDNA elements. Sequencing-based methods, including AmpliconArchitect, AmpliconReconstructor, Circle_Finder, and Circle-Map, deduce the ecDNA structures from whole-genome sequencing (WGS) data. In contrast, Circle-Seq and CRISPR-CATCH provide targeted ecDNA analysis with enhanced

resolution. Furthermore, the ecTag method facilitates the visualization of ecDNA in alive cells through labeling the ecDNA-specific sequences with guide RNAs (gRNAs) and fluorescent markers.”

We believe that this addition enhances the manuscript's completeness by providing readers with a comprehensive overview of the diverse methodologies available for ecDNA characterization. Once again, we express our gratitude to the reviewer for their insightful feedback, which has enriched the content of our study.

The authors say: “.....We selected auPRC (area under precision-recall curve) as the primary evaluation metric on gene-level prediction due to the extreme class imbalance. As expected, the copy number, rather than other molecular profiles, exhibits moderate predictive ability (auPRC = 0.595) (Extended Data Fig. 2b). The data indicate that copy number is the most practical feature among the molecular signatures being investigated for ecDNA prediction...”

This formulation should be formulated more carefully, like The data may indicate that... normally ecDNAs are present at high copy number, which facilitates detection through sequencing-based approaches, as the number of copies scales linearly with the number of derived sequencing reads,supporting the sentence of the authors

Response: We appreciate the reviewer's suggestion and understand the concern regarding the description of our results. We have revised the corresponding content as follows: “We selected auPRC (area under precision-recall curve) as the primary evaluation metric on gene-level prediction due to the extreme class imbalance. As expected, the copy number exhibited moderate predictive ability (auPRC = 0.595) rather than other molecular profiles (Extended Data Fig. 2b). The data might imply that ecDNAs are commonly found with elevated copy counts, a trait advantageous for sequencing-based detection strategies, given that the quantity of copies varies proportionally alongside the quantity of resulting sequencing reads. Consequently, gene copy number emerges as a practical molecular characteristic for predicting ecDNA amplifications.”

REVIEWER COMMENTS

Reviewer #1 (Remarks to the Author):

The reviewers have addressed the critiques through further analyses and this has greatly strengthened the paper. It is a valuable contribution to the field.

Reviewer #2 (Remarks to the Author):

I thank the authors for their very thorough and well-written response to my concerns, and I thank them for the additional analysis they performed. The revisions they have performed have helped strengthen and clarify the manuscript.

This reviewer raised the point that discrimination of BFB with GCAP may be challenging and the additional analysis the authors performed specifically for this reviewer seems to confirm that point. However, no specific mention of that challenge has made its way into the manuscript - only more indirect, general references to the potential for false-positives or false-negatives. It will be important for the audience of the paper to understand that some of these missed calls are systematic in nature based on the biology of the focal amplification - not random effects happening uniformly across the data. In other words, BFB focal amplifications appear to suffer the same difficulties in discrimination using WES data as they do with WGS data.

Reviewer #1 raised a very good point that more extensive validation was needed. The manuscript authors responded by including more samples they performed FISH on. Undoubtedly, the GCAP method has high sensitivity, as it relies predominantly on easy-to-detect copy number amplifications to detect ecDNA. However, this reviewer argues that the validation served in response to reviewer #1 is not sufficient as it only contains samples the authors claim are ecDNA+ (positive control). HSR-like ecDNA-negative focal amplifications must also be included as a negative control so we can better understand to what extent the method is able to discriminate ecDNA from any other mode of focal amplification.

Concerningly, upon examining the new validation FISH data included in the revised manuscript (Supplementary File 2), the last 6 images which contain ERBB2 amplifications do not appear to this reviewer to be extrachromosomal. Instead, all of these appear to be multi-focal homogeneously-staining region (HSR) amplifications. How did the authors reach the conclusion that those 6 ERBB2-amplified samples were on ecDNA? Text in Supplemental File 2 claims that these FISH blobs exhibit "donut-shaped morphology", which seems strange, as the size of the FISH blob is not at all consistent with an ecDNA given the level of magnification in the images. If the authors proceed with calling these samples ecDNA+, they should back it up with another technology (e.g. WGS) because it does not at all appear to be obvious those 6 ERBB2 samples are ecDNA+ given the FISH images. The same issue exists with the image Extended Data Figure 3 (lower panel), where there appears to be one or two amplification clusters per cell - a hallmark of HSR.

Reviewer #3 (Remarks to the Author):

All my concerns were addressed. There was a very comprehensive and careful processing.

Nature Communications Manuscript #NCOMMS-23-29647

Responses to Reviewers' comments

We express our gratitude to all the reviewers for acknowledging the importance of our study and expressing satisfaction with the revised manuscript. In the following section, we specifically address the concerns raised by Reviewer #2, in a point-to-point way.

Reviewers' comments:

Reviewer #2:

I thank the authors for their very thorough and well-written response to my concerns, and I thank them for the additional analysis they performed. The revisions they have performed have helped strengthen and clarify the manuscript.

General response: We extend our sincere appreciation to the reviewer for dedicating time to reevaluate our manuscript. We are grateful for the reviewer's acknowledgment of the improvements made in our revised manuscript.

This reviewer raised the point that discrimination of BFB with GCAP may be challenging and the additional analysis the authors performed specifically for this reviewer seems to confirm that point. However, no specific mention of that challenge has made its way into the manuscript - only more indirect, general references to the potential for false-positives or false-negatives. It will be important for the audience of the paper to understand that some of these missed calls are systematic in nature based on the biology of the focal amplification - not random effects happening uniformly across the data. In other words, BFB focal amplifications appear to suffer the same difficulties in discrimination using WES data as they do with WGS data.

Response: We appreciate the reviewer's insightful comments. In the previous version of our manuscript, we outlined the constraints of GCAP with WES data, highlighting the inherent limitations stemming from the utilization of copy number features for predictions. This was in

response to our comprehensive examination of false positives and false negatives, as detailed in our previous response letter. However, we recognize that our summary in the discussion has lacked sufficient depth regarding to its biological insights. Here, to address the concern, we have refined and supplemented the relevant discussion section as follows.

"It is crucial to highlight that elevated genomic copy numbers are observed in both non-ecDNA focal amplifications (e.g., BFB) and ecDNA amplification events. Given the pivotal role of copy number as a predictive feature, non-ecDNA focal amplifications with increased copy numbers may be prone to false positives (e.g., in cancer cell lines HCC827 and SH10TC). Simultaneously, low-copy-number ecDNAs that have not undergone significant positive selective pressures may result in false negatives (e.g., in cancer cell lines FU97 and KYSE180). The challenges faced by BFB focal amplifications in discrimination using WES data parallel those observed with WGS data."

Reviewer #1 raised a very good point that more extensive validation was needed. The manuscript authors responded by including more samples they performed FISH on. Undoubtedly, the GCAP method has high sensitivity, as it relies predominantly on easy-to-detect copy number amplifications to detect ecDNA. However, this reviewer argues that the validation served in response to reviewer #1 is not sufficient as it only contains samples the authors claim are ecDNA+ (positive control). HSR-like ecDNA-negative focal amplifications must also be included as a negative control so we can better understand to what extent the method is able to discriminate ecDNA from any other mode of focal amplification.

Response: We thank the Reviewer for raising additional questions and concerns about the validation in our study based on the issues posted by Reviewer #1.

In addressing the concern raised by the reviewer regarding the perceived inadequacy of negative controls leading to insufficient validation of GCAP, we find it essential to provide clarification. Our study incorporates a substantial amount of negative control data. For instance, we conducted a comparison between GCAP and AmpliconArchitect, and both cell batch 1 and cell batch 2 include data with ecDNA-negative instances. Taking the data from cell batch 2 as an example (https://zenodo.org/records/8373312/files/Batch2_AA_summary.xlsx), AmpliconArchitect and GCAP jointly identified seven cell lines (NUGC3, NUGC4, OCUM1, AGS, IM95, 2313287, SNU1)

as ecDNA-negative, among which five (NUGC3, NUGC4, OCUM1, AGS, SNU1) were identified as non-ecDNA focal amplifications by AmpliconArchitect. Furthermore, our pan-cancer analysis revealed substantial distribution differences in molecular features associated with ecDNA between circular and noncircular patient groups across diverse cohorts such as TCGA, PCAWG, and SYSUCC CRC. This provides robust population-level validation of the distinguishability between the two amplification types. Additionally, in our previous revision, we directly validated GCAP's ability to detect ecDNA through Circle-Seq results from 11 cell lines, addressing concerns raised by Reviewer #1.

Regarding the suggestion to include HSR-like focal amplifications as controls, it is crucial to underscore the impracticality of such an approach due to the conceptual vagueness of HSR-like amplification in genomics. HSR-like amplification is a phenotype observed in cell cytogenetics and is primarily utilized to describe DNA segments with uniform staining intensity after G banding. Extensive reviews and supporting experimental evidence already highlight the intricate relationships between BFB, HSR, and ecDNA (e.g., **REF 1-3**). Early reports have suggested a potential relationship or interconvertibility between these two forms of amplification, as illustrated in a screenshot (**Response Figure 1**) from the review by Bafna et al. This interconvertibility is proposed to occur through BFB cycles. Deshpande et al. explored and supported the common origin and interconversion of ecDNA and HSR during the development of the AmpliconArchitect method (Figure 4 of **REF 4**). In our FISH experiment targeting *MYC*, we detected HSR-like amplification signals in the established ecDNA+ cell line PC3 (**REF 5**), and a cell in metaphase is highlighted (**Response Figure 2**). In our FISH experiment targeting *ERBB2* in the MKN7 cell line, identified as ecDNA+ by GCAP, AmpliconArchitect (AA), and Circle-Seq, we once again observed certain amplification patterns reminiscent of HSR-like structures (**Response Figure 3**). Luebeck et al.'s study (Figure 2 of **REF 6**) also provides a compelling example of reconstructing ecDNA from HSR-like amplifications. Therefore, observing HSR-like amplification signals in FISH does not exclude the presence of ecDNA; instead, it to some extent suggests the existence of ecDNA (see also **Response Table 1**). This perspective may elucidate the absence of any mention of HSR by Kim et al. in their use of AmpliconArchitect for classifying amplification modes on cancer WGS data (**REF 7**). Most recently, under the revision of this work, a new tool for detecting ecDNA from ATAC-seq data was reported by Cheng et al. (**REF 8**), wherein ecDNA and HSR were tiered together

both in concepts and approach. Combined, HSR-like focal amplifications conflict with ecDNA-negative instances and are not suitable negative control.

We appreciate the reviewer's keen observations. These diverse biological concepts and descriptions represent our understanding of cancer genome amplification from various perspectives. We anticipate researchers leveraging a combination of diverse methods and technologies to delve into the nuances of conceptual ambiguities and conduct thorough investigations on focal amplification modes.

[redacted]

Response Figure 1. Snapshot adopted from figure 5 of **REF 1** for your easy reference (open access at <https://www.annualreviews.org/doi/full/10.1146/annurev-genom-120821-100535>).

Response Figure 2. Fluorescence in situ hybridization (FISH) image depicting cancer cell line PC3 with targeted *MYC* gene (in red) and adjacent chromosomal locus (in green). A cell in metaphase stage is highlighted. Scale bar: 10 micrometers.

Response Figure 3. Fluorescence in situ hybridization (FISH) images depicting cell heterogeneity of cancer cell line MKN7 with targeted *ERBB2* gene (in red) and adjacent chromosomal locus (in green). Two selected cells are shown. Scale bar: 5 or 10 micrometers.

REF list:

1. Bafna, Vineet, and Paul S. Mischel. "Extrachromosomal DNA in cancer." *Annual Review of Genomics and Human Genetics* 23 (2022): 29-52
2. Ilić, Mila, et al. "Life of double minutes: generation, maintenance, and elimination." *Chromosoma* 131.3 (2022): 107-125
3. Storlazzi, Clelia Tiziana, et al. "Gene amplification as double minutes or homogeneously staining regions in solid tumors: origin and structure." *Genome research* 20.9 (2010): 1198-1206
4. Hung, King L., et al. "ecDNA hubs drive cooperative intermolecular oncogene expression." *Nature* 600.7890 (2021): 731-736.
5. Deshpande, Viraj, et al. "Exploring the landscape of focal amplifications in cancer using AmpliconArchitect." *Nature communications* 10.1 (2019): 392
6. Luebeck, Jens, et al. "AmpliconReconstructor integrates NGS and optical mapping to resolve the complex structures of focal amplifications." *Nature communications* 11.1 (2020): 4374
7. Kim, Hoon, et al. "Extrachromosomal DNA is associated with oncogene amplification and poor outcome across multiple cancers." *Nature genetics* 52.9 (2020): 891-897
8. Cheng, Hansen, et al. "ATACamp: a tool for detecting ecDNA/HSRs from bulk and single-cell ATAC-seq data." *BMC genomics* 24.1 (2023): 1-10.

Concerningly, upon examining the new validation FISH data included in the revised manuscript (Supplementary File 2), the last 6 images which contain ERBB2 amplifications do not appear to this reviewer to be extrachromosomal. Instead, all of these appear to be multi-focal homogeneously-staining region (HSR) amplifications. How did the authors reach the conclusion that those 6 ERBB2-amplified samples were on ecDNA? Text in Supplemental File 2 claims that these FISH blobs exhibit "donut-shaped morphology", which seems strange, as the size of the FISH blob is not at all consistent with an ecDNA given the level of magnification in the images. If the authors proceed with calling these samples ecDNA+, they should back it up with another technology (e.g. WGS) because it does not at all appear to be obvious those 6 ERBB2 samples are ecDNA+ given the FISH images. The same issue exists with the image Extended Data Figure 3 (lower panel), where there appears to be one or two amplification clusters per cell - a hallmark of

HSR.

Response: We sincerely appreciate the examinations and thoughtful insights provided by the reviewer. It is crucial to emphasize that our interpretation of FISH results from clinical samples, particularly regarding *ERBB2* amplifications, is undertaken with utmost caution. In the manuscript, we expressed our consideration of two potential amplification patterns indicative of ecDNA: diffuse amplified signals, representing a classical extrachromosomal DNA amplification pattern (more confident), and locally clustered amplified signals, suggesting a potential gene exchange process involving ecDNAs and chromosome homogeneously staining regions (HSRs) (less confident). The Supplementary File 2 provides extra data of these two amplification patterns as supplementary evidence supporting our findings. To further enhance precision and alleviate potential misinterpretations, we have added clarifications when describing the amplification patterns in our manuscript and the supplemental text, underscoring our cautious approach to result interpretation: "It becomes evident that nearly all images display either diffuse gene amplification signals (**more confident**) and/or clustered gene amplification signals (**less confident**) highlighted within cyan boxes." Similarly, concerning the issue raised about Extended Data Figure 3 (lower panel), we observed the presence of multiple independent *ERBB2* amplification signals away from the adjacent chromosomal locus chr17 (**Response Figure 4**). We believe that the evidence of both diffuse and locally clustered amplified signals in CRC1057 rationalizes its inclusion as supplementary validation data. Additionally, detailed explanations regarding the relationship between ecDNA and HSR have been provided in the response above; hence, further elaboration will not be reiterated here.

Response Figure 4. Snapshot of fluorescence in situ hybridization (FISH) probing *ERBB2* within sample of CRC1057, part of Extended Data Figure 3 (lower panel).

Moreover, regarding the reference to "donut-shaped morphology", we wish to clarify that this description was made in the context of observing a peculiar amplification signal in the CRC211 patient sample (**Response Figure 5**). We found the observation intriguing and documented it, explicitly stating, "It appears that some clustered amplification signals exhibit a donut-shaped morphology." However, we did not intend to imply that such signals represent circular DNA molecules or serve as conclusive evidence of ecDNA existence. To avoid potential misconceptions, we have removed this description in the revised submission.

Response Figure 5. Snapshot of fluorescence in situ hybridization (FISH) probing *ERBB2* within sample of CRC211.

Finally, taking into consideration that *ERBB2* amplification typically exhibits a pattern of locally clustered gene amplification signals in the FISH results, which provides less confidence in validating both the presence of ecDNA and *ERBB2* amplification on the ecDNA, the reviewer suggested that "all of these appear to be multi-focal homogeneously-staining region (HSR) amplifications." In response to this concern, we conducted additional sample collection and performed whole-genome sequencing (WGS) followed by AmpliconArchitect analysis on the six *ERBB2*-amplified samples (Note: CRC1057, displayed in Extended Data Figure 3 (lower panel), had no available samples for sequencing). The AmpliconArchitect analysis on WGS data revealed that five of these six samples exhibited cyclic DNA amplicons (**Response Table 1 and Response Figure 6**, also included in Supplemental File 2). Among these, three were classified as ecDNA+ (**Response Table 1**). Notably, none of the six *ERBB2* amplifications were classified as linear amplification. Besides, CRC938, the only sample predicted by GCAP to harbor both the ecDNA cargo gene *MYC* and *ERBB2*, was validated through the AmpliconArchitect analysis (**Response Table 1**). Combined, our extended analysis not only addresses the reduced confidence in FISH results but also reinforces the robustness of GCAP in analyzing clinical samples.

Response Table 1. AmpliconArchitect result summary. All detected amplicons are included.

Sample	amplicon_decomposition_class	Feature ID	Classification	Oncogenes
CRC211	Cyclic	CRC211_amplicon1_BFB_1	BFB	['CDC6', 'CSF3', 'ERBB2', 'RARA', 'THRA']
CRC259	Cyclic	CRC259_amplicon1_ecDNA_1	ecDNA	['ERBB2']
CRC259	Cyclic	CRC259_amplicon2_ecDNA_1	ecDNA	[]
CRC259	Cyclic	CRC259_amplicon2_BFB_1	BFB	[]
CRC350	Cyclic	CRC350_amplicon1_ecDNA_1	ecDNA	['CDK12', 'ERBB2']
CRC634	Complex non-cyclic	CRC634_amplicon1_Complex non-cyclic_1	Complex non-cyclic	['ERBB2']
CRC938	Cyclic	CRC938_amplicon1_ecDNA_1	ecDNA	['ERBB2', 'MYC', 'PVT1']
CRC983	Cyclic	CRC983_amplicon1_BFB_1	BFB	['CDK12', 'ERBB2']

NOTE: full columns at https://zenodo.org/records/10212116/files/AA_summary_table_of_6_erb2_ffpe_samples.xlsx.

Response Figure 6. Radar-style plots of amplicon classification strengths.

In conclusion, we extend our gratitude to the reviewer #2 for their keen observations, professional advice, opinions, and guidance. These contributions have significantly enhanced the clarity and readability of the manuscript, effectively presenting our novel methodology in a scientifically sound manner. The reviewer's insights have not only addressed potential issues but

also greatly benefited both our team and the manuscript.

REVIEWERS' COMMENTS

Reviewer #2 (Remarks to the Author):

I thank the authors for their thorough response, and for the careful examination of the data added in their prior revision.

The updates to the discussion section and the WGS validation provided to their imaging data appropriately satisfy my concerns.